# Characterization of a chemical modulation reactor (CMR) for the measurement of atmospheric concentrations of hydroxyl radicals with a laser-induced fluorescence instrument

Changmin Cho[1], Andreas Hofzumahaus[1], Hendrik Fuchs[1], Hans-Peter Dorn[1], Marvin Glowania[1], Frank Holland[1], Franz Rohrer[1], Vaishali Vardhan[1], Astrid Kiendler-Scharr[1], Andreas Wahner[1] and Anna Novelli[1]

[1]Forschungszentrum Jülich, Institute for Energy and Climate Research: Troposphere (IEK-8), 52425 Jülich, Germany

*Correspondence to*: Andreas Hofzumahaus (a.hofzumahaus@fz-juelich.de) and Anna Novelli (a.novelli@fz-juelich.de)

**Abstract.** Precise and accurate hydroxyl radical (OH) measurements are essential to investigate mechanisms for oxidation and transformation of trace gases and processes leading to the formation of secondary pollutants like ozone ($O_3$) in the troposphere. Laser induced fluorescence (LIF) is a widely used technique for the measurement of ambient OH radicals and was used for the majority of field campaigns and chamber experiments. Recently, most LIF instruments in use for atmospheric measurements of OH radicals introduced chemical modulation to separate the ambient OH radical concentration from possible interferences by chemically removing ambient OH radicals before they enter the detection cell (Mao et al., 2012; Novelli et al., 2014a). In this study, we describe the application, and characterization of a chemical modulation reactor (CMR) applied to the Forschungszentrum Jülich LIF (FZJ-LIF) instrument in use at the atmospheric simulation chamber SAPHIR. Besides dedicated experiments in synthetic air, the new technique was extensively tested during the year-round Jülich Atmospheric Chemistry Project (JULIAC) campaign, in which ambient air was continuously flowed into the SAPHIR chamber. It allowed for performing OH measurement comparisons with Differential Optical Absorption Spectroscopy (DOAS) and investigation of interferences in a large variety of chemical and meteorological conditions. Good agreement was obtained in the LIF-DOAS intercomparison within instrumental accuracies (18 % for LIF, 6.5 % for DOAS) which confirms that the new chemical modulation system of the FZJ-LIF instrument is suitable for measurement of interference-free OH concentrations under the conditions of the JULIAC campaign (rural environment). Known interferences from $O_3$+$H_2O$ and the nitrate radical ($NO_3$) were quantified with the CMR in synthetic air in the chamber and found to be $3.0 \times 10^5$ cm$^{-3}$ and $0.6 \times 10^5$ cm$^{-3}$, respectively, for typical ambient air conditions ($O_3$ = 50 ppbv, $H_2O$ = 1%, $NO_3$ = 10 pptv). The interferences measured in ambient air during the JULIAC campaign in the summer season showed a median diurnal variation with a median maximum value of $0.9 \times 10^6$ cm$^{-3}$ during daytime and a median minimum value of $0.4 \times 10^6$ cm$^{-3}$ at night. The highest interference of $2 \times 10^6$ cm$^{-3}$ occurred in a heat wave from 22 – 29 August, when the air temperature and ozone increased to 40°C and 100 ppbv, respectively. All observed interferences could be fully explained by the known $O_3$ + $H_2O$ interference, which is routinely corrected in FZJ-LIF measurements when no chemical modulation is applied. No evidence for an unexplained interference was found during the JULIAC campaign.

A chemical model of the CMR was developed and applied to estimate the possible perturbation of the OH transmission and scavenging efficiency by reactive atmospheric trace gases. These can remove OH by gas phase reactions in the CMR, or produce OH by non-photolytic reactions, most importantly by the reaction of ambient $HO_2$ with NO. The interfering processes become relevant at high atmospheric OH reactivities. For the conditions of the JULIAC campaign with OH reactivities below 20 s$^{-1}$, the influence on the determination of ambient OH concentrations was small (on average: 2 %). However, in environments with high OH reactivities, such as in a rain forest or megacity, the expected perturbation in the currently used chemical modulation reactor could be large (more than a factor of 2). Such perturbations need to be carefully investigated and corrected for the proper evaluation of OH concentrations when applying chemical scavenging. This implies that chemical

modulation, which was developed to eliminate interferences in ambient OH measurements, itself can be subject to interferences that depend on ambient atmospheric conditions.

## 1 Introduction

The hydroxyl radical (OH) plays an important role in tropospheric photochemistry as the main daytime oxidant of trace gases (e.g. volatile organic compounds (VOCs), nitrogen oxides ($NO_x$=NO+$NO_2$), and carbon monoxide (CO)) removing primary pollutants while contributing to ozone ($O_3$) and particle formation (Levy, 1971; Carslaw et al., 2010; Li et al., 2019). In the lower troposphere, OH is mainly produced by two types of atmospheric reactions. First, solar photolysis of trace gases (most importantly ozone and nitrous acid (HONO)),

$$O_3 + h\nu\ (< 340nm)\ \rightarrow\ O(^1D) + O_2 \qquad\qquad\qquad R1$$

$$O(^1D)\ +\ H_2O\qquad \rightarrow\ 2OH \qquad\qquad\qquad\qquad R2$$

$$HONO + h\nu\ (< 400nm)\ \rightarrow\ OH + NO \qquad\qquad\qquad R3$$

Second, by reaction of NO with hydroperoxy radicals ($HO_2$)

$$HO_2\ +\ NO\ \rightarrow\ OH + NO_2 \qquad\qquad\qquad\qquad\qquad R4$$

which are produced by photolysis of formaldehyde (HCHO) and as intermediates in the OH-initiated degradation of CO and VOCs or the ozonolysis of alkenes. A large number of measurements of OH radical concentrations in ambient air have been conducted in various environments so far (Stone et al., 2012; Feiner et al., 2016; Tan et al., 2017; Mallik et al., 2018; Tan et al., 2018; Whalley et al., 2018; Ma et al., 2019; Lew et al., 2020). Some of these field studies included the detection of peroxy radicals, trace gases, and measurements of the OH reactivity ($k_{OH}$), the inverse of OH lifetime, which helped with investigation of the atmospheric OH radical budget. Good agreement between measured OH radicals and calculations using chemical box modelling that represent the current understanding of tropospheric chemistry is often found at high NO concentrations (> 1 parts per billion (ppbv)), which favour non-photolytic OH production by reaction R4. In pristine environments (NO < 1 ppbv), where OH reactivity is dominated by biogenic VOCs, in particular isoprene, a large discrepancy, up to a factor of 10, was found with the measured OH radical concentrations being higher than calculated. This discrepancy suggested that an unknown atmospheric OH-formation process is missing from the chemical mechanisms (Tan et al., 2001; Lelieveld et al., 2008; Hofzumahaus et al., 2009; Kubistin et al., 2010; Whalley et al., 2011; Rohrer et al., 2014).

Theoretical (Peeters et al., 2009; da Silva et al., 2010; Peeters and Müller, 2010; Peeters et al., 2014; Wang et al., 2018; Møller et al., 2019) and experimental studies (Crounse et al., 2011; Berndt, 2012; Crounse et al., 2012; Wolfe et al., 2012; Fuchs et al., 2013; Fuchs et al., 2014; Teng et al., 2017; Fuchs et al., 2018; Berndt et al., 2019; Novelli et al., 2020) have hypothesized and found significant regeneration of OH radicals from unimolecular reactions of organic peroxy radicals ($RO_2$, R = organic group) originating from the oxidation of biogenic VOCs such as isoprene, methacrolein, and methyl vinyl ketone (MVK). Although the inclusion of these new reaction paths in atmospheric chemical models increases the predicted concentration of OH considerably for some conditions, for example by a factor of three over tropical forests (Novelli et al., 2020), it is often not enough to explain the high OH concentrations observed in field experiments in areas characterized by high isoprene emissions (Stone et al., 2011; Lu et al., 2012; Stone et al., 2012; Fuchs et al., 2013; Lu et al., 2013; Wolfe et al., 2014; Feiner et al., 2016; Lew et al., 2020).

An explanation for the remaining discrepancies between measured and modeled OH radical concentrations in high isoprene and low NO environments could be interferences in OH radical measurements. In most of the field campaigns described above, the OH radical was measured by gas expansion of ambient air into a low-pressure volume, where the radicals are detected by laser-induced fluorescence (LIF) at 308nm, except for the study by Wolfe et al. (2014) where the OH radical was measured by chemical ionization mass spectrometers (CIMS). Laser wavelength modulation is usually applied to distinguish the OH fluorescence from laser excited background signals. Previous studies have investigated and reported possible interferences in

the detection of ambient OH by LIF, which originate from the formation of OH inside the instrument. Such internal OH can be formed by laser photolysis of ozone in the presence of water (reactions R1 + R2) (Holland et al., 2003; Ren et al., 2004) and acetone (Ren et al., 2004; Fuchs et al., 2016), ozonolysis of alkenes (Ren et al., 2004; Novelli et al., 2014b; Fuchs et al., 2016; Rickly and Stevens, 2018), and by unknown reactions of nitrate radicals ($NO_3$) (Fuchs et al., 2016). Most of these

interfering species are thought not to play a role at ambient concentrations with one exception. Ozone photolysis in humid air may be relevant, but can be corrected for based on laboratory characterization experiments (Holland et al., 2003).

In accordance with a recommendation from the International HOx workshop 2015 (Hofzumahaus and Heard, 2016), the majority of LIF instruments now apply chemical modulation in order to correct possible interferences. Chemical modulation in instruments measuring OH, which is also commonly used by CIMS which can measure OH by addition of $SO_2$ in form of

$HSO_4^-$ by ionization with $NO_3^-$ (Eisele and Tanner, 1991; Tanner et al., 1997; Berresheim et al., 2000; Mauldin et al., 2010), involves periodic scavenging of the ambient OH by addition of a reactant (propane or hexafluoropropene) before the air enters the detection cell. Any remaining OH fluorescence signal can then be attributed to OH produced inside the detection cell (Mao et al., 2012; Novelli et al., 2014a; Tan et al., 2017; Rickly and Stevens, 2018; Tan et al., 2018; Tan et al., 2019; Woodward-Massey et al., 2020) (Table 1). The difference between the signal without and with the scavenger provides interference-free

ambient OH signals which are then used to calculate OH radical concentrations by means of a calibration.

Several LIF instruments applying chemical modulation (MPIC, PSU, and IU in Table 1) have shown relatively large OH interferences which seem to depend on the chemical conditions of the sampled air as well as on specific instrument characteristics (e.g. inlet size and shape or whether the detection cell is single-path or multiple path) and/or from the combination of both. After subtraction of the interference, a significantly improved agreement was found between measured

OH concentrations and chemical box model predictions (Mao et al., 2012; Feiner et al., 2016; Mallik et al., 2018; Lew et al., 2020). In forest environments, measured interferences contributed 40 - 80 % to the total signal in daytime and 50 -100 % in nighttime (Mao et al., 2012; Hens et al., 2014; Novelli et al., 2014a; Feiner et al., 2016; Lew et al., 2020). Smaller contributions of 20 - 40 % to the total signal were found in daytime coastal (Mallik et al., 2018), rural (Novelli et al., 2014a), and urban (Brune et al., 2016; Griffith et al., 2016) environments, but nighttime observation were similar to those found in forested

environments. Some studies suggest that the interferences may be partly caused by the dissociation of stabilized Criegee intermediates (SCIs) that are produced by ozonolysis of alkenes (Novelli et al., 2014b; Novelli et al., 2017; Rickly and Stevens, 2018). However, other LIF instruments using chemical modulation (PKU and Leeds in Table 1) showed insignificant unexplained interferences after the well-quantified photolytic ozone interference had been subtracted (Tan et al., 2017; Tan et al., 2018; Tan et al., 2019; Woodward-Massey et al., 2020). Although these studies were not performed in forested

environments with large BVOC emissions as some of the studies where a large interference was observed, during the summer campaign in Wangdu (Tan et al., 2017) and Beijing (Woodward-Massey et al., 2020), high isoprene concentrations (up to 3 ppbv and 7.9 ppbv, respectively) did not seem to perturb the level of the interference. These field campaign results are consistent with the results of laboratory and/or chamber studies (Fuchs et al., 2016; Woodward-Massey et al., 2020) which found insignificant interference from the ozonolysis of BVOCs under atmospheric conditions.  It remains an open question

which chemical species and processes produce unexplained OH interferences in some of the LIF instruments and how technical design and operating conditions of the instruments influence the magnitude of the unexplained interferences. Chemical modulation seems to be an appropriate way to minimize such measurement artefacts. However, as will be discussed in this paper, the accurate evaluation of chemical modulation measurements can be challenging and may have systematic errors in the measured OH concentration, because chemical reactions of atmospheric trace gases can disturb the efficiency of chemical

modulation.

In this study, the chemical modulation reactor (CMR) used in the LIF instrument of Forschungszentrum Jülich (FZJ) is introduced, described, and characterized. The characterization includes laboratory tests and chamber experiments. Furthermore, a theoretical model is developed that provides estimates of the possible influence of ambient atmospheric

conditions on the chemical modulation efficiency and can possibly be used for the correction of such influences. Measurements of OH radicals using the CMR LIF technique are compared to measurements by Differential Optical Absorption Spectroscopy (DOAS) in synthetic and ambient air inside the atmospheric simulation chamber SAPHIR. Most experiments were part of the Jülich Atmospheric Chemistry Project (JULIAC) in which ambient air from 50 meters height was continuously drawn through the chamber for one month in each season of the year. This allowed the investigation of possible interference signals for a large range of chemical and meteorological conditions in a rural environment.

## 2 Methods

### 2.1 Forschungszentrum Jülich LIF OH instrument with chemical modulation reactor (FZJ-LIF-CMR)

The LIF instrument for the measurement of OH, $HO_2$ and $RO_2$ radicals in use at Forschungszentrum Jülich has been extensively described in several previous studies (Holland et al., 2003; Fuchs et al., 2008; Fuchs et al., 2012a). Briefly, the measurement of OH radicals is achieved by sampling ambient air through a 0.4 mm pinhole nozzle into a low-pressure (4 hPa) detection cell. The OH radical is resonantly excited by a single-path UV laser pulse (308 nm) on a single rovibronic transition ($Q_1(3)$) of the $A^2\Sigma^+$ - $X^2\Pi$ (0,0) electronic system using a narrow-bandwidth tunable dye laser. The resulting OH fluorescence is detected by a micro-channel plate photomultiplier (MCP) connected to a gated photon counting system. To distinguish the OH fluorescence from non-resonant laser-excited background, the excitation wavelength is modulated on and off the peak of the OH absorption line (Hofzumahaus et al., 1996). The laser system is operated at a high repetition rate of 8.5 kHz and the MCP signals are integrated over 25 s and 10 s in the on- and off- resonance mode, respectively. A complete cycle takes 45 s. The measured OH concentration resulting from the wavelength modulation is hereafter called $OH_{WAVE}$ following the nomenclature introduced by Mao et al. (2012).

$$[OH]_{WAVE} = \frac{S_{WAVE}}{C_{OH}} = \frac{1}{C_{OH}} \times (S_{ON} - S_{OFF}) \tag{1}$$

Here, $S_{WAVE}$ is the OH radical fluorescence signal normalized to the UV laser power. It is obtained from the difference between $S_{ON}$ and $S_{OFF}$, the on- and off-resonance signals, respectively. $C_{OH}$ represents the OH radical detection sensitivity, which is obtained during the calibration of the instrument. The instrument is calibrated for OH radicals by illuminating humidified synthetic air (Linde, $N_2$ and $O_2$ purity > 99.99990) by a mercury lamp thereby producing a known amount of OH radicals (approximately $5.0 \times 10^9$ $cm^{-3}$ at 0.8 % water vapor mixing ratio) by photolysis of water vapor at 185 nm. The intensity of the light is determined by actinometry and it is measured by a calibrated photo-diode. The 1σ accuracy of the calibration is 10 % (Holland et al., 2003). The detailed design of the radical source and the calibration procedure are described in previous studies (Holland et al., 2003; Fuchs et al., 2011).

Hydroxyl radicals originating not from ambient air but formed within the detection cell will also be detected if they are excited by the laser in the detection volume. To quantify the interference signal from internally generated OH radicals, a chemical modulation reactor (CMR) has been developed and mounted in front of the OH detection cell of both the LIF system permanently mounted at the SAPHIR chamber (in this study) and the LIF system used in field campaigns (Tan et al., 2017; Tan et al., 2018; Tan et al., 2019). The CMR outline is shown in Figure 1 and instrumental characteristics are listed in Table 1. The reactor consists of a 79 mm long PTFE Teflon tube with 10 mm internal diameter that is mounted in an aluminum body. Two stainless steel injectors (1/8" OD tubes with 50μm ID) are located 50 mm above the nozzle pinhole pointing to the center of the CMR tube. For typical measurement conditions, 20.6 slpm (1 slpm = 1 L min⁻¹ at 1 atm, 20°C) of ambient air are sampled. A constant flow of 500 sccm (1 sccm = 1 cm³ min⁻¹ at 1 atm, 20°C) of nitrogen ($N_2$, purity > 99.9990 %) or a mixture of nitrogen and the OH scavenger (Airliquid, Propane, purity>99.95%, 5.0±0.1% mixture in nitrogen, purity>99.999%) regulated by mass flow controllers is injected into the main stream. When propane is added, the resulting propane mixing-ratio

downstream of the injectors is typically 19 ppmv. The concentration of propane was selected to scavenge OH radicals efficiently (90 – 95 %) in the CMR but to avoid scavenging OH radicals in the detection cell of the instrument (see section 3.2). Propane is used as an OH scavenger because it does not photolyse at the wavelength of the laser and does not react with other oxidants (e.g. $O_3$) which could produce OH radicals in the CMR. When switching from the mixture of scavenger and nitrogen to pure nitrogen, the injection lines are flushed for 15 s at a higher flow rate of 700 sccm to remove any residual scavenger molecule from the lines. The measurement time with addition of propane is 135 s, followed by another period of 135 s without propane injection. A complete cycle therefore takes 270 s. The flow rate behind the injectors is 21.1 slpm. It is controlled by a flow controller (Bronkhorst, low $\Delta$-p) connected to a membrane pump which removes 20 slpm of excess air, while the OH measurement cell samples 1.1 slpm.

Without added scavenger ($N_2$ mode of operation), the OH signal obtained from an on-off-resonance measurement cycle is

$$S_{OH}^{N2} = S_{OH} + S_i \tag{2}$$

It contains the signal $S_{OH}$ from ambient OH radicals which pass the CMR and reach the inlet nozzle of the detection cell, and potentially an interference signal $S_i$ from OH radicals that are produced inside the cell (internal OH). When the scavenger is added (sc mode of operation), a large percentage (typically 96 %) of atmospheric OH radicals is removed by reaction with propane. The OH signal in the scavenger mode is

$$S_{OH}^{SC} = \alpha \, S_{OH} + S_i \tag{3}$$

It consists of the OH signal from the residual ambient OH, which is reduced by a factor $\alpha$ due to scavenging, and $S_i$. The residual factor $\alpha$ has to be determined experimentally (see Section 3), in order to calculate $S_{OH}$ and $S_i$ from a set of measurements in the scavenging and nitrogen modes.

$$S_{OH} = \frac{S_{OH}^{N2} - S_{OH}^{sc}}{1 - \alpha} \tag{4}$$

$$S_i = \frac{S_{OH}^{sc} - \alpha \, S_{OH}^{N2}}{1 - \alpha} \tag{5}$$

The conversion of $S_{OH}$ into an interference-free ambient OH concentration, called $[OH]_{CHEM}$, requires calibration.

$$[OH]_{CHEM} = \frac{1}{C_{OH} \beta_{N2}} \, S_{OH} \tag{6}$$

In addition to the detection sensitivity $C_{OH}$ of the OH detection cell without the CMR, the OH transmission of the CMR in the $N_2$ mode ($\beta_{N2}$) needs to be known. In the present system, the transmission is reduced due to wall loss reactions and has a typical value of 64 %. The quantities $C_{OH}$ and $\beta_{N2}$ have to be calibrated either separately or together ($C_{OH} \times \beta_{N2}$) by calibrating the OH cell with the CMR in place (see Section 3).

The interference OH signal $S_i$ can be converted into an equivalent ambient OH concentration in analogy to Eq. (6).

$$[OH]_i = \frac{1}{C_{OH} \beta_{N2}} \, S_i \tag{7}$$

It should be noted that the application of the CMR relies on the assumption that interfering chemical species, which produce OH inside the instrument, are not affected by switching between $N_2$ and scavenger injection in the CMR. Furthermore, it is assumed that internal OH is not scavenged by propane inside the OH detection cell. The latter assumption was confirmed for this instrument (see Section 3).

## 2.2 Additional instrumentation

A large suite of additional instruments was available for experimental studies in the SAPHIR chamber. Measurements of interest are listed in Table 2 with 1σ accuracies and precisions. The calibration-free differential optical absorption spectroscopy (DOAS) instrument (White cell, absorption path length: 2.2 km) provided an absolute reference for OH radical measurements with a 1σ accuracy of 6.5 % (Hausmann et al., 1997; Schlosser et al., 2007; Schlosser et al., 2009). The pressure and temperature dependence of the OH absorption cross section has been discussed in detail in the study by Dorn et al. (1995).

Within the natural variance of the atmospheric pressure and a temperature interval of ±20K around room temperature the OH cross section changes less than 2%. The LIF instrument for OH concentration measurement includes two more measurement cells for the detection of $HO_2$ and $RO_2$ radicals. In the second LIF cell ($HO_X$ cell), $HO_2$ is indirectly measured by chemical conversion to OH by NO (Linde, mixture of 1 % NO in nitrogen (purity > 99.999 %)) (Fuchs et al., 2011). During the JULIAC
experiments, NO ($2.5 \times 10^{13} cm^{-3}$) was used for minimizing a possible interference from specific $RO_2$ radicals as suggested in Fuchs et al. (2011). OH reactivity ($k_{OH}$) was measured by a laser-photolysis laser-induced fluorescence instrument (LP-LIF) (Lou et al., 2010; Fuchs et al., 2017b). Photolysis frequencies were calculated from spectral actinic flux densities measured by a spectroradiometer (Bohn et al., 2005; Bohn and Zilken, 2005). In addition, $NO_3$ radical concentrations were monitored by a custom-built cavity ring-down spectroscopy (FZJ-CRDS) instrument; nitrous acid (HONO) by a long-path absorption
photometer (LOPAP) (Kleffmann et al., 2006; Li et al., 2014); water vapour mixing ratio by a cavity ring-down spectroscopy instrument (CRDS, Picarro); nitric oxide (NO) and nitrogen dioxide ($NO_2$) by a chemiluminescence instrument with a photolytic converter (CL, Eco Physics); and $O_3$ by UV absorption instruments (Ansyco-41M and Thermo scientific-49I) that agree within 5%.

## 2.3 Atmospheric simulation chamber SAPHIR

Comprehensive tests of the FZJ-LIF-CMR instrument with synthetic (Section 3) and ambient air (Section 4) were performed in the atmosphere simulation chamber SAPHIR on the campus of Forschungszentrum Jülich, Germany as the instrument is permanently mounted at the SAPHIR chamber. A detailed description of the chamber has been presented in previous publications (Bohn and Zilken, 2005; Rohrer et al., 2005; Fuchs et al., 2010b). Briefly, the chamber has a cylindrical shape (length: 18 m, diameter: 5 m, volume: 270 m$^3$) and is made of a double Teflon (FEP) film, which has a light transmission
greater than 0.8 over the complete solar spectral range (Bohn and Zilken, 2005). A slight overpressure is maintained inside the chamber and clean nitrogen is used to flush the volume between the inner and outer FEP films to prevent contamination from outside the chamber. Two fans are mounted inside the chamber to ensure a homogeneous mixing of trace gases.
The SAPHIR chamber is an ideal tool to test and characterize instruments for the measurement of atmospheric trace gases, as it was shown for the measurements of OH (Schlosser et al., 2007; Schlosser et al., 2009; Fuchs et al., 2012a), $HO_2$ (Fuchs et
al., 2010b), $NO_2$ (Fuchs et al., 2010a), $NO_3$ (Dorn et al., 2013), $N_2O_5$ (Fuchs et al., 2012b), and OH reactivity (Fuchs et al., 2017a). In this study, the chamber was used for experiments with synthetic air (Linde, purity: > 99.99990 %), to test the CMR for known interferences from ozone/water and $NO_3$ radicals (Section 3), and with ambient air (Section 4).

## 2.4 JULIAC campaign

The CMR system was further used for OH measurements in SAPHIR within the Jülich Atmospheric Chemistry Project (JULIAC) in 2019 during which ambient air was continuously flowed through the chamber. The campus is surrounded by a mixed deciduous forest and is located close to the small town of Jülich. Therefore, ambient air is expected to be influenced by both anthropogenic and biogenic emissions. The JULIAC campaign consisted of four intensive phases, each carried out within one season (Table 3). It was designed to examine the seasonal and diurnal cycle of atmospheric oxidants in a mixed
environment and the role of oxidants in the degradation of VOCs, NOx, and formation/ageing of particles. Ambient air was sampled at 50m height through a SilcoNert® coated inlet line (104 mm ID) at a flow rate of 660 m$^3$ h$^{-1}$. Around 250 m$^3$ h$^{-1}$ of this air flow were directed towards the SAPHIR chamber. This new inlet system for SAPHIR enable all instruments to measure the same air composition without the perturbation of steady state conditions by local emitters. The inlet line was gently heated (+1-2°C above ambient) to avoid water vapor condensation in the line. A cyclone was positioned upstream of the SAPHIR
chamber to remove particles with a diameter larger than 10μm. The residence time of the ambient air in the chamber was about one hour giving enough time for radicals and short-lived trace gases to reach a new steady state and smoothing atmospheric

variability. Daytime-averaged meteorological conditions and trace gas concentrations during each JULIAC intensive phase are summarized in Table 3 and nighttime-averaged data are shown in Table S2. Daytime is defined by the condition that the $NO_2$ photolysis frequency is greater than $0.5 \times 10^{-4}$ s$^{-1}$. In Table 3 and S2, the total OH reactivity of non-methane VOCs ($k_{VOC}$) is calculated by subtracting the contribution of CO, $CH_4$, $O_3$, NO, and $NO_2$ from the measured OH reactivity. The tables show that a large range of chemical and meteorological conditions was encountered during the four different JULIAC campaigns allowing the investigation of possible interferences in the OH radical measurement for different scenarios (e.g., summer/winter, high/low NO and $O_3$, high/low humidity) in a rural environment.

## 3 Characterisation and test of the CMR for clean air conditions

This section describes the characterization and test of the CMR for conditions where the measured air is very clean. This condition was simulated by performing experiments in synthetic air.

### 3.1 CMR transmission

The loss of OH radicals in the CMR and the scavenging efficiency were determined in laboratory experiments using the photolytic radical source which is also used for absolute calibration of the instrument. As usual, OH radicals are produced by 185 nm photolysis of water vapour in a flow of synthetic air at 1 atm and room temperature, where the concentration of formed OH is determined from the humidity, volume flow, and the intensity of the 185 nm radiation. The latter is observed by a phototube which can be traced back to a measurement of ozone which is produced in the same gas flow by 185 nm photolysis of oxygen (for details see Fuchs et al. (2011)). The radical source was operated with a total flow of 24 slpm which overflowed the inlet of the CMR tube. The length of the laminar flow tube (18.7 mm ID with a frit at the top, Reynolds number Re = 1920) of the radical source is 20 cm, resulting in plug flow condition that ensures a uniform distribution of OH and $O_3$ in the calibration gas. This condition was confirmed by ozone measurements with a modified chemiluminescence instrument (50 pptv limit-of-detection) showing that the ozone concentration in the central flow was the same within a few percent compared to the mean ozone concentration leaving the radical source.

Several tests were performed to characterize heterogeneous OH losses at surfaces in the CMR. Three reactive regions can be distinguished (Figure 2): the entrance section above the injectors, the injector tubes, and the reaction section downstream of the injectors. The flow in the entrance section is in the transition regime between laminar and turbulent (Reynolds number, Re = 2800), but turbulence is further increased by the injectors which protrude approximately 4mm into the flow tube. In order to quantify potential OH loss on the surface of the injectors, the stainless steel injectors were replaced by Teflon tubes of the same geometry for a test. Within measurement precision (±3%), no difference in the detected OH was found when the material of the injectors was changed. Since Teflon and stainless steel have orders of magnitude different surface reaction probabilities for radicals (Rozhenshtein et al., 1985), the result suggests that OH loss at the injector surfaces is negligible regardless of its material.

Since the injector tubes do not cause OH losses, the OH transmission in the $N_2$ mode ($\beta_{N2}$) can be described as the product of transmissions of the entrance section ($\beta^e$) and reaction section ($\beta^r_{N2}$) (see Figure 2).

$$\beta_{N2} = \beta^e \times \beta^r_{N2} \tag{8}$$

As a next step, the values of $\beta^e$ and $\beta^r_{N2}$ were determined experimentally using clean air as carrier gas for OH. The clean air condition ($k_{OH} = 0$) will be indicated by the superscript (0) in the following. First, the OH transmission of the CMR tube without built-in injectors was determined by measuring OH from the radical source with and without the CMR mounted on the OH cell. A transmission $\beta^0_{tube}$ of 0.81 ± 0.02 was determined for a flow rate of 21.1 slpm. Assuming that the OH loss is caused by only wall reactions and follows first order kinetics, a rate coefficient $k_w = 11.8$ s$^{-1}$ is calculated from

$$\beta_{tube}^0 = \exp(-k_w \Delta t) \tag{9}$$

where $\Delta t$ is the transit time through the tube. The rate coefficient, $k_w$, that was obtained can be used to estimate the transmission of the entrance section ($\beta^{e,0}$),

$$\beta^{e,0} = \exp(-k_w^e \Delta t_e) \tag{10}$$

which has a transit time $\Delta t_e = 6.6$ ms, yielding $\beta^{e,0} = 0.92$.

With built-in 1/8" injectors and nitrogen injection, an OH transmission of $\beta_{N2}^0 = 0.64 \pm 0.03$ was measured for the total CMR. The wall loss of OH radicals for a flow rate in the CMR between 12 and 21.1 slpm was investigated (Figure S1). When the flow rate increases, it has two opposing effects. First, it decreases the residence time in the CMR leading to a steady increase of the OH transmission from about 60% at 12 slpm to 65% at 15 slpm. Second, an increasing flow rate causes more turbulence and faster radial transport to the wall. This is the likely reason for flattening of the transmission curve at flow rates above 15 slpm. Further increase of the flow rate more than 21 slpm does not improve the OH transmission in synthetic air. To minimize the possible effect of secondary chemistry, the CMR was operated with the fastest flow rate achievable (21.1 slpm), which corresponds to a transmission a transmission $\beta_{N2}^0$ of 0.64.

Using the experimental values of $\beta_{N2}^0$ and $\beta^{e,0}$, the relationship (Equation 8) yields $\beta_{N2}^{r,0} = 0.69$ as transmission of the reaction section. With a transit time $\Delta t_r = 11.2$ ms, a wall loss rate coefficient $k_w^r = 33$ s$^{-1}$ is obtained.

The same procedure was used to determine also the CMR transmission for HO$_2$ radicals, which provides information needed to calculate OH production in the CMR by Reaction R4 (Section 4). In this case, the calibration source was operated with added CO (300 ppmv) in order to convert all OH to HO$_2$ in the calibration gas (Fuchs et al., 2011). In order to detect HO$_2$, the CMR was mounted on a HOx detection cell. The measured transmissions and wall loss rate coefficients for HO$_2$ are listed in Table 4 together with the results for OH.

The rate coefficients for HO$_2$ are a factor of two smaller than for OH, because HO$_2$ is less reactive. If the wall loss was limited by the reactive collision frequency at the wall surfaces, a larger difference would be expected, because the reaction probability $\gamma$ on Teflon surfaces is about a factor of ten smaller for HO$_2$ compared to OH (Rozhenshtein et al., 1985). The rate coefficient $k_s$ for wall loss due to reactive surface collisions can be calculated for a cylindrical tube with radius R by

$$k_s = \gamma \frac{\bar{u}}{2R} \tag{11}$$

if transport in the gas-phase presents no resistance (Zasypkin et al., 1997). For OH with a mean molecular velocity $\bar{u}$ of 610 m/s (298 K) and $\gamma = 2.5\times10^{-3}$ for Teflon surfaces (Rozhenshtein et al., 1985), the value of $k_s$ is 152 s$^{-1}$. The measured rate coefficients for OH loss ($k_w$) (Table 4) are an order of magnitude smaller, indicating that OH loss in the CMR tube is limited by turbulent transport to the walls, which is about three times faster in the reaction section compared to the CMR entrance region.

## 3.2 Scavenging efficiency

The same radical source operated with clean synthetic air was also used for the characterization test of scavenging efficiency. When propane is injected into the CMR, the chemistry in the reaction section changes. When no propane is injected (N$_2$ mode), OH is only lost by wall reaction.

$$\beta_{N2}^{r,0} = \exp(-k_w^r \Delta t_r) \tag{12}$$

In the scavenger mode, additional OH loss occurs by the gas-phase reaction with propane. The OH transmission is then given by

$$\beta_{sc}^{r,0} = \exp(-[k_w^r + k_{sc}]\Delta t_r) \tag{13}$$

where $k_{sc}$ is the pseudo first-order rate constant for reaction between OH and propane.

$$k_{sc} = k_{OH+propane}[propane] \tag{14}$$

Here, $k_{OH+propane}$ is the bimolecular rate constant for scavenging, which has a value of $1.1 \times 10^{-12}$ cm$^3$ s$^{-1}$ at 298 K (Atkinson et al., 2006).

The fraction of ambient OH remaining that subsequently enters the measurement cell downstream of the CMR when the CMR is operated in the scavenging mode is the ratio of $\beta_{sc}^r$ and $\beta_{N2}^r$.

$$\alpha = \frac{\beta_{sc}^r}{\beta_{N2}^r} \qquad (15)$$

which in the specific case of synthetic air ($k_{OH} = 0$) becomes

$$\alpha^0 = \frac{\beta_{sc}^{r,0}}{\beta_{N2}^{r,0}} = \exp(-k_{sc} \Delta t_r) \qquad (16)$$

$\alpha^0$ was experimentally determined for OH in synthetic air from the radical source by calculating the ratio of OH signals measured with and without scavenger. The measured ratio is denoted $\alpha^0$, where the superscript indicates that the sampled air does not contain any additional OH reactants ($k_{OH} = 0$). Note that the value of the residual factor '$\alpha$' is different for ambient air ($k_{OH} \neq 0$) as is discussed in Section 4. Figure 3a and 3b show the measured dependence of $\alpha^0$ on the added amount of propane for different instrumental conditions. For small amounts of propane, $\alpha^0$ decreases nearly exponentially as expected from Equation 16, whose derivation assumes homogenous mixing of the scavenger. At higher scavenger concentrations, however, $\alpha^0$ levels off into a weak dependency and decreases slower than expected from an exponential decay. This behavior indicates inhomogeneous mixing of propane in the CMR flow.

Figure 3a also shows that the use of 1/8" injectors leads to improved mixing compared to 1/16" injectors. The likely reason is that the 1/8" injectors produce a larger flow resistance within the CMR and produce higher turbulence in the flow because of their larger surface. The disadvantage of the 1/8" injectors is a smaller CMR transmission, $\beta_{N2}^0 = 0.64$, compared to a value of 0.75 for the 1/16" injectors. Thus, the faster mixing of propane using the 1/8" injectors goes along with a higher OH wall loss.

Figure 3b shows that the carrier gas (N$_2$) flow rate has only a minor influence on $\alpha^0$, when the injector flow rate is changed between 100 sccm and 500 sccm. In order to ensure fast exchange of the gases in the injector lines, the higher flow rate of 500 sccm was chosen for routine operation. For the majority of the results shown in this study, the 1/8" injectors were used to maximize the mixing in the CMR. The 1/16" injectors were only tested during part of the JULIAC experiments (Table S1).

Ideally, $\alpha$ would be zero. In that case, the OH signal in the scavenger mode would be identical with the interference signal ($S_i = S_{OH}^{sc}$) and the ambient OH signal would simply be given by the signal difference of both modes ($S_{OH} = S_{OH}^{N2} - S_{OH}^{sc}$). Higher propane concentrations should result in smaller $\alpha$ values, but Figure 3a shows that further increase of the propane concentration beyond 20 ppmv gives only small improvements in $\alpha$. In order to avoid OH scavenging in the detection cell by too high propane concentrations that would lead to an underestimation of the interference signal, the propane concentration in the CMR was chosen as 19 ppmv for routine operation. Note that 19 ppmv would be the nominal value if the injected propane is homogeneously mixed into the total CMR flow. For this condition, 96 % of OH is scavenged ($\alpha^0 = 0.042$). This value corresponds to an effective scavenging rate coefficient $k_{sc}^{eff} = 283$ s$^{-1}$ in Equation 16. It is a factor of 1.8 smaller than calculated for complete homogenous mixing ($k_{sc} = 513$ s$^{-1}$).

A plausible explanation for the weaker than expected dependence of $\alpha$ on the propane concentration is incomplete mixing of the injected scavenger. A minor fraction f of a few percent of the total CMR flow containing little or no scavenger would explain the observed dependence of $\alpha^0$ in Figure 3a. The fraction with no scavenger ($k_{sc}' = 0$) can be imagined as gas filaments with $\alpha_1(k_{sc}' = 0) = 1$ embedded in the turbulent flow. The bulk flow, i.e. the major part (1-f), would contain all the injected propane well mixed, with $\alpha_2$ ($k_{sc}'' = k_{OH+propane}[Propane]$). The resulting $\alpha^0$ of the total CMR flow would then be

$$\alpha^0 = (1-f)\,\alpha_2\left(k_{sc}''\right) + f\,\alpha_1(k_{sc}' = 0)$$
$$= (1-f)\,\alpha_2\left(k_{sc}''\right) + f \qquad (17)$$

Using Equation 16 and 17, we obtain

$$\alpha^0 = (1 - f)\exp\left(k_{sc}^{''} \Delta t_r\right) + f \tag{18}$$

This is a simple model for the case of inhomogeneous mixing with an assumed value f = 0.04, which provides a better description of the observed dependence of $\alpha$ on the propane concentration than Equation 16 (Figure 3a, c). In this model, the calculated reactivity $k_{sc}^{''}$ is 534 $s^{-1}$, which is larger than the $k_{sc}$ for complete homogeneous mixing as the injected propane is assumed to be mixed in a smaller volume flow (96% of the total volume flow).

### 3.3 Scavenging of internal OH

As mentioned above, the concentration of propane is chosen not to scavenge any OH radical in the detection cell. Calculations suggest that less than 1% of OH radicals are scavenged in the detection cell at 19 ppmv of propane because of the low pressure (4 hPa) and the short residence time (3 ms) in the detection cell. A laboratory test to quantify internal OH removal in the detection cell was proposed by Woodward-Massey et al. (2020) and was also done in this study. As employed in the $HO_2$ transmission tests, the radical source was operated as a pure $HO_2$ source. The CMR was mounted on the HOx detection cell, which used 10 times higher NO concentration than during typical operation for ambient $HO_2$ concentration measurements by an injection of 8 sccm of 10% NO (Linde NO 10%, mixture in nitrogen) to achieve a high $HO_2$ to OH radical conversion efficiency of 85 %. The $HO_2$ radical concentration provided by the radical source is not affected by the propane injection in the CMR, but the $HO_2$ is converted to OH in the detection cell. If a significant amount of OH radicals was scavenged by propane inside the detection cell, a difference between the measured signals with and without propane injection in the CMR would be observed. Different propane concentrations (10 to 75 ppmv) were tested during the experiment (Figure 3d). A small amount of (3±2) % of internal OH was scavenged when 19 ppmv of propane was applied. An increase of internal removal up to a value of (5±3) % was observed for the highest propane mixing ratio (75 ppmv). The internal scavenging value of (3±2) % for the operation conditions in this work applies to OH radicals, which are internally formed and exposed to reaction with propane on a similar time scale as in the $HO_2$ to OH conversion (ca. 2 ms) in the test experiment. OH that is produced photolytically in the laser beam has a much shorter residence time (ca. 0.1ms) before it is removed by the fast gas flow and will therefore be much less affected by internal scavenging. As discussed by Woodward-Massey et al. (2020), internal OH that is immediately formed after the sampled air has passed the inlet nozzle would have a longer residence time, which could be larger by up to a factor of two compared to the $HO_2$ conversion experiment. Even in this case, the internal scavenging would be a small effect and is therefore considered negligible.

### 3.4 OH detection sensitivity and calibration

As mentioned in Section 2.1, the conversion of interference-free ambient OH signals $S_{OH}$ into ambient concentrations requires the experimental determination of the parameters $C_{OH}$ and $\beta_{N2}$ (Equation 6). The combined detection sensitivity $C_{OH} \times \beta_{N2}^0$ of the measurement system with mounted CMR was calibrated with the radical source, which provided a known OH concentration in humidified synthetic air. The superscript (0) in $\beta_{N2}^0$ indicates that the calibration was done in synthetic air ($k_{OH} = 0$). The calibration was independently tested by an OH measurement comparison against the OH DOAS instrument which provides an absolute reference for OH concentrations in the SAPHIR chamber. The chamber was filled with clean synthetic air. Only water vapor (1.3%) was injected and the chamber roof was opened allowing for the photolytic formation of OH radicals from HONO, which is photochemically produced at the chamber walls (Rohrer et al., 2005). Figure 4 shows the OH radical concentrations measured by DOAS and LIF-CMR ([OH]$_{CHEM}$). The difference between the measured time series is on average (0.29 ± 0.9) $\times 10^6$ $cm^{-3}$, which is less than 5% of the average measured OH concentrations between 08:30 and 12:00. Thus, the instruments agreed well within the combined 1σ accuracies of the LIF-CMR calibration (± 18%) and DOAS (± 6.5%), which confirms the LIF calibration. A linear regression (not shown) which considers the precision of both instruments using the algorithm Fitexy

between LIF-CMR and DOAS yields a slope of 0.97. The good agreement gives confidence in the applied chemical modulation technique under the operational conditions, and in the laboratory-determined parameters $\alpha^0$ and $C_{OH} \times \beta_{N2}^0$ used for data evaluation.

The implementation of chemical modulation in the FZJ-LIF instrument has specific disadvantages compared to the operation of the LIF OH detection without the CMR. It requires a longer measurement time, because half of the time is spent for the measurement of interferences in the scavenger mode. In the current work, measurement of ambient OH ($N_2$ mode, 135s) contains three on-off resonance cycles which yield three ambient OH data points (45 s). The subsequent scavenger mode (135s) takes again three on-off resonance cycles. Another disadvantage of the CMR is the reduction of the OH detection sensitivity by a factor of 1.6 due to wall loss in the CMR, yielding a $1\sigma$ limit-of-detection (LOD) of $7 \times 10^5$ cm$^{-3}$ (Table 2). Note that this LOD value applies to a single OH data point of 45 s duration.

Accuracy and precision of the OH measurements are generally worsened, because [OH]$_{CHEM}$ requires more experimental parameters ($C_{OH}$, $\alpha$, $\beta_{N2}$) for evaluation than OH$_{WAVE}$, as well as more measurement modes, each of which contribute additional noise. The $1\sigma$ accuracy of OH measurements depends on the error of the radical source (10%) and the reproducibility of the experimental determination of $C_{OH} \times \beta_{N2}^0$ and $\alpha^0$. During the one-year JULIAC campaign (Section 4), the reproducibilities were 15% and 10%, respectively, resulting in a total $1\sigma$ accuracy of $\pm18$ %. Additional uncertainties arise when air pollutants influence the chemistry in the CMR (see Section 4).

### 3.5 Interference tests

The main purpose of the CMR is to discriminate the signal of ambient OH radicals from signals of known and unknown interferences. Two types of known interferences (from ozone photolysis and NO$_3$ radicals) were re-investigated in this work by making use of the chemical modulation technique. These tests were done in synthetic air to avoid potential interferences from other sources.

### 3.5.1 Interference from ozone photolysis

It is well-known that internal OH can be produced in the 308 nm laser beam of LIF systems by photolysis of O$_3$ in humid air according to reactions R1 + R2 (Holland et al., 2003).

The interference is proportional to the concentrations of ozone and water vapour, and laser power:

$$OH_{Interference, O_3 + H_2O} \propto [O_3] \times [H_2O] \times \text{Laser Power} \tag{19}$$

In the past, OH$_{WAVE}$ measurements by the FZJ-LIF instrument were routinely corrected for this interference based on laboratory characterization. Similar tests were repeated here without and with the CMR for a range of conditions (Figure 5). Mixing ratios of ozone and water vapour were varied up to 450 ppbv and 1.8 %, respectively. Laser power was modulated between 10 and 20 mW. A laboratory test was performed with the calibration source as gas supply with a total flow of 11 slpm and photolysis lamp switched off. Ozone was added by a home-built ozone generator and measured in the excess air with a UV photometery (Ansyco). Relative humidity was monitored by a humidity sensor (HMT 333, Vaisala). In two additional experiments in the SAPHIR chamber, water vapour and ozone were added in synthetic air in dark conditions (chamber roof closed) and their concentrations were measured by the instruments listed in Table 2. The OH interferences observed in the laboratory and SAPHIR experiments scale linearly with the product of ozone and water vapour mixing ratios as expected (Equation 19, Figure 5). All three experiments with and without the CMR agree with each other within 15 %. On average, the interference is equivalent to an OH concentration of $(3.4 \pm 0.3) \times 10^5$ cm$^{-3}$ per 50 ppbv of O$_3$ and 1 % water mixing ratio, which is at the limit-of-detection for the LIF instrument without the CMR, and a factor of two below the limit-of-detection with the CMR. The value of the interference agrees well with earlier determinations for the FZJ-LIF instrument with reported

values of $(2.7 \pm 0.8) \times 10^5$ cm$^{-3}$ (Holland et al., 1998) and $(3.2 \pm 0.8) \times 10^5$ cm$^{-3}$ (Holland et al., 2003) for the same conditions.

### 3.5.2 Interference from NO₃

Fuchs et al. (2016) have reported an OH interference from NO₃ radicals by an unknown mechanism producing OH in the FZJ-LIF instrument. The observed interference was independent of water vapour in the gas phase and independent of laser power. The reported interference signal in the presence of 10 pptv NO₃ was equivalent to an atmospheric OH concentration of $1.1 \times 10^5$ cm$^{-3}$. In the present work, the interference from NO₃ was re-determined applying chemical modulation. The experiments were performed in the SAPHIR chamber using thermal decomposition of N$_2$O$_5$ added from a condensed source which produces NO₃ radicals. Figure 6 shows a linear increase of the measured OH interference with increasing NO₃ concentration, which is measured by a cavity ringdown instrument similar to the instrument described in Fuchs et al. (2009) and Wagner et al. (2011). A linear regression analysis yields a value that is equivalent to an OH concentration of $5.8 \times 10^4$ cm$^{-3}$ per 10 pptv NO₃, which is negligible for typical ambient concentrations of NO₃. The value is roughly 2 times smaller than the result of Fuchs et al. (2016). The reason for the discrepancy between the two tests is unclear. One reason could be an additional NO₃ heterogeneous loss to the wall of the CMR, like HO₂ and OH loss, when the CMR is applied. As the origin of the NO₃ radical interference in our LIF instrument is unknown, it cannot be ruled out that other experimental conditions affected the interfering signal. In any case, the small interference from NO₃ is negligible for OH measurements for the FZJ-LIF instrument, regardless of whether chemical modulation is used or not. A similar conclusion was drawn from experiments by Woodward-Massey et al. (2020) for the LIF instrument used by the University of Leeds (UK).

## 4 Chemical modulation measurements in ambient air

This section describes the application of chemical modulation in a real atmosphere with numerous atmospheric pollutants. Section 4.1 analyses the potential influence of atmospheric chemical conditions on the performance of the chemical modulation system and Section 4.2 shows test results from the JULIAC field campaign.

### 4.1. Sensitivity study of the influence of ambient air conditions on chemical modulation

### 4.1.1 Kinetic model

When polluted air with sufficiently high OH reactivity ($k_{OH}$) is sampled by the CMR, the reaction of OH with atmospheric reactants competes with the OH loss by wall reactions and scavenging inside the CMR. Therefore, an influence of $k_{OH}$ on the OH transmission and scavenging efficiency is expected. In addition, some reactions that produce OH in the ambient air continue when the air enters the CMR. A simple kinetic model is used here to quantify these influences.

In the atmosphere, the ambient concentration of short-lived OH radicals is in steady state and can be calculated from the total OH production rate and OH reactivity.

$$[OH]_a = \frac{P_{hv} + P_d}{k_{OH}} \tag{20}$$

Here, $P_{hv}$ represents the photolytic OH production rate, which is dominated in the lower troposphere by the photolysis of O$_3$ and HONO (R1-R3). $P_d$ denotes non-photolytic (dark) OH production, which includes the reaction of HO₂ radicals with NO (Reaction R4), ozonolysis of alkenes and, for example, isomerization reactions of RO₂ leading directly to OH formation.

When atmospheric OH radicals enter the inner darkness of the chemical modulation reactor, photolysis will stop, but chemical reactions of OH with atmospheric reactants and dark OH production will continue, as long as the reactants that produce or

destroy OH are not lost in the CMR tube. In many atmospheric environments, the reaction between $HO_2$ and NO (Reaction R4) with a reaction rate constant of $k_{HO2+NO} = 8.1 \times 10^{-12}$ cm$^3$ s$^{-1}$ at 298 K (Atkinson et al., 2004) is the dominant non-photolytic source of atmospheric OH. It can be assumed that the OH production rate from Reaction 4 continues inside the CMR at the same rate as in the ambient atmosphere, because concentrations of $HO_2$ and NO do not significantly change. $HO_2$ is relatively short-lived, but during the short transit time of 18 ms through the reactor, its conversion to OH is small for typical conditions. Even high ambient NO mixing ratios of 30ppbv lead only to a conversion of $HO_2$ to OH of less than 10%. Furthermore, $HO_2$ loss will be partly compensated by $HO_2$ production from the reaction of atmospheric $RO_2$ with NO. Also, the loss of $HO_2$ by wall reactions is small (18%) (Section 3.1). NO has an even longer chemical lifetime than $HO_2$, which is usually determined by the reaction of NO with $O_3$ and peroxy radicals ($HO_2$ and $RO_2$).

As a result of the ongoing chemical reactions, ambient OH will relax in the CMR to a new steady state with concentration $[OH]'$ and chemical lifetime $\tau'_{OH}$.

$$[OH]' = \frac{P_d}{k_{OH} + k_w + k_{sc}} \tag{21}$$

$$\tau'_{OH} = \frac{1}{k_{OH} + k_w + k_{sc}} \tag{22}$$

The relaxation follows first order kinetics and the reaction time t in the reactor determines the degree of relaxation. The general time law for such a transition from an initial concentration $[OH]_0$ (t = 0) to a final concentration $[OH]'$ is given by

$$[OH](t) = [OH]_0 + \{[OH]' - [OH]_0\} \left[1 - \exp\left(-\frac{t}{\tau'_{OH}}\right)\right] \tag{23}$$

Using this expression, an OH transmission $\beta$ can be defined for a tube section with transit time $\Delta t$.

$$\beta = \frac{[OH](\Delta t)}{[OH]_0} \tag{24}$$

The initial concentration ($[OH]_0$) can be expressed as reduced ambient OH concentration $c\,[OH]_a$, with a reduction factor c = 1.0 for the entrance section and c = $\beta^e$ for the reaction section (cf., Figure 2). Making use of the relationships in Equation 20, 21 and 22, the transmission can be generally written as

$$\beta\,(c, k_{OH}, k_w, k_{sc}, \Delta t) = 1 + \left\{\frac{1}{c} \frac{P_d}{P_{hv}+P_d} \times \frac{k_{OH}}{k_{OH} + k_w + k_{sc}} - 1\right\} [1 - \exp(-[k_{OH} + k_w + k_{sc}]\,\Delta t)] \tag{25}$$

More specifically, we obtain for the entrance and reaction sections in the $N_2$ and scavenger mode the following transmissions:

$$\beta^e = \beta\,(c = 1, k_{OH}, k_w^e, k_{sc} = 0, \Delta t_e) \tag{26}$$

$$\beta^r_{N2} = \beta\,(c = \beta^e, k_{OH}, k_w^r, k_{sc} = 0, \Delta t_r) \tag{27}$$

$$\beta^r_{sc} = \beta\,(c = \beta^e, k_{OH}, k_w^r, k_{sc}, \Delta t_r) \tag{28}$$

These formulas can be inserted in Equations 8 and 15 to calculate $\beta_{N2}$ and $\alpha$ as a function of $k_{OH}$, respectively. Note that for the case of $k_{OH} = 0$, Equations 26 - 28 become identical with Equations 10, 12, and 13, respectively.

### 4.1.2 Scavenging model for the case of homogeneous mixing

The kinetic model from Section 4.1 is used to estimate the influence of the ambient OH reactivity and dark OH production on the determination of interferences and ambient OH concentrations. The OH wall loss rate coefficients, $k_w^e$ and $k_w^r$, are taken from Table 4. The reaction times are $\Delta t_e = 6.6$ ms and $\Delta t_r = 11.2$ ms. We assume homogeneous mixing of the propane in the scavenging mode applying an effective rate coefficient $k_{sc}^{eff} = 283$ s$^{-1}$ that was calculated from the experimental $\alpha^0 = 0.042$ (Section 3.1).

As explained in Section 2.1, the extraction of $S_{OH}$ and $S_i$ from the measured OH signals in the scavenging and nitrogen modes (Equation 4 and 5) requires the knowledge of the remaining fraction $\alpha$ of ambient OH in the scavenging mode. Figure 7a presents modelled $\alpha$ as a function of atmospheric OH reactivity for different ratios of dark-to-total OH production rates

$P_d/(P_{hv}+P_d)$. The dependence is shown for reactivities from 0 to 100 s$^{-1}$, a range that is typical for the lower troposphere (e.g., Lou et al. (2010)), and for $P_d/(P_{hv}+P_d)$ ratios between 0 and 1. The parameter $\alpha$ shows a considerable variation for these conditions with the largest effect at high OH reactivities. For $k_{OH} = 100$ s$^{-1}$, $\alpha$ varies from 0.042 to 0.3 for variation of $P_d/(P_{hv}+P_d)$ from 0 to 1. The latter case with $P_d \gg P_{hv}$ is typical at nighttime. In the special case of $P_d = 0$, when no OH is produced inside the CMR, the residual factor $\alpha$ remains constant (same as the experimental $\alpha^0 = 0.042$) and becomes independent of $k_{OH}$. In this case, $k_{OH}$ adds to the total first-order loss rates and cancels out in Equation 15, yielding Equation 16. Low $P_d$ values would be expected, for example, in a low NO environment, where the reaction of HO$_2$ with NO does not play a significant role. Under atmospheric conditions when the production $P_d$ exceeds the photolytic OH production, the remaining fraction of ambient OH reaching the nozzle increases due to the ongoing dark OH production in the CMR. The sensitivity to $k_{OH}$ increases with $P_d/(P_{hv}+P_d)$ and becomes largest at nighttime, when ambient OH production is controlled by dark reactions, only.

The sensitivity of the calculated signals $S_{OH}$ (Equation 4) and $S_i$ (Equation 5) to variations of $\alpha$ (Figure 7a) can be seen in Figure 7c and d, respectively. A biased signal $S_{OH}^*$ would be obtained if the influence of $k_{OH}$ on the chemical modulation was neglected.

$$\frac{S_{OH}^*}{S_{OH}} = \frac{1-\alpha}{1-\alpha^0} \tag{29}$$

The corresponding biased interference signal $S_i^*$ is similarly:

$$\frac{S_i^* - S_i}{S_{OH}} = 1 - \frac{S_{OH}^*}{S_{OH}} \tag{30}$$

Figure 7d demonstrates that the interference signal would be systematically overdetermined, while the ambient OH signal would be correspondingly underestimated (Figure 7c). This effect is largest at night, where up to 25% of the true ambient OH signal would be wrongly assigned as interference at $k_{OH} = 100$ s$^{-1}$. The error remains generally small ($< 5\%$) for $P_d/(P_{hv}+P_d)$ ratios in the range between $0 - 1$, if OH reactivities stay below 10 s$^{-1}$. More generally, the influence of $k_{OH}$ on $\alpha$ is small when $k_{OH}$ is much lower than the scavenger reactivity (here 283 s$^{-1}$) and therefore introduces only a small perturbation in the CMR chemistry.

### 4.1.3 Scavenging model for the case of inhomogeneous mixing

The model results in Figure 7a, c, and d assume homogeneous mixing of the scavenger with an effective reactivity of $k_{sc}^{eff} = 283$ s$^{-1}$. As an alternative model, Equation 17 can be used to simulate inhomogeneous mixing. As explained in Section 3.2, this model assumes that a small fraction $f = 0.04$ of air contains little or no scavenger, and therefore does not contribute to chemical modulation ($\alpha_1 = 1$). The main bulk of the gas flow, i.e. part $(1-f)$ of the CMR flow, is assumed to contain the injected propane well mixed with a scavenger reactivity of $k_{sc}^{''} = 534$ s$^{-1}$. Equations 16, 27 and 28 are then applied to calculate the $k_{OH}$ dependence of $\alpha_2$ in Equation 17 to determine $\alpha$. The result presented in Figure 8a is very similar to the homogeneous mixing case shown in Figure 7a. Again, the residual factor $\alpha$ is constant (0.042) and independent of $k_{OH}$ for $P_d = 0$. In this model, the constant offset represents ambient OH that was not scavenged in the minor fraction of air with no scavenger. In the bulk flow, however, essentially all OH is depleted by the high scavenger reactivity of 534 s$^{-1}$. For $P_d > 0$, $\alpha$ increases with $k_{OH}$ as in the case of homogeneous mixing. The maximum sensitivity to $k_{OH}$ is again obtained for a $P_d/(P_{hv}+P_d)$ ratio of 1. The main difference between the results for the homogeneous and inhomogeneous mixing is a slightly weaker sensitivity to $k_{OH}$ in the case of inhomogeneous mixing due to the higher $k_{sc}$ value in the bulk flow. Thus, $\alpha$ varies from 0.042 to 0.22 at $k_{OH} = 100$ s$^{-1}$ for inhomogeneous mixing, whereas the span is slightly larger reaching 0.3 in the homogeneously mixed case. As the results for $\alpha$ are very similar, also the ratio $S_{OH}^*/S_{OH}$ gives comparable results (Figure 7c and Figure 8b).

### 4.1.4 Transmission model

The conversion of the signals $S_{OH}$ and $S_i$ to concentrations requires the knowledge of $\beta_{N2}$ (Equation 6 and 7). As the calibration of our LIF-CMR instrument determines $C_{OH} \times \beta_{N2}^0$ (Section 3.4), Equation 6 can be rewritten as

$$[OH]_{CHEM} = \frac{1}{C_{OH}\beta_{N2}^0} \times \frac{\beta_{N2}^0}{\beta_{N2}} \times S_{OH} \qquad (31)$$

Here, the normalized function $\beta_{N2}/\beta_{N2}^0$ contains the $k_{OH}$ dependence which is shown in Figure 7b. $\beta_{N2}/\beta_{N2}^0$ is much more sensitive to OH reactivity from OH reactants contained in the ambient air than the remaining fraction of OH in the scavenger mode $\alpha$. Without scavenger, the atmospheric OH reactivity competes only with wall loss, for which the rate is a similar magnitude as $k_{OH}$. In an environment with little NO, i.e., for $P_d/(P_{hv}+P_d) = 0$, $k_{OH}$ has to be added to the wall loss rate. The normalized ratio $\beta_{N2}/\beta_{N2}^0$ decreases exponentially with $-k_{OH}(\Delta t_e + \Delta t_r)$ in this case. Under this condition, the normalized

transmission decreases by a factor of 6 when $k_{OH}$ reaches 100 s$^{-1}$. Without taking this effect into account, the calculated ambient OH concentration would be extremely underestimated in an environment with very little NO. However, when the OH reactivity remains below 10 s$^{-1}$, OH would be underestimated by less than 15%.

When the value of $P_d/(P_{hv}+P_d)$ ratio exceeds 0.7, the $k_{OH}$ dependence of the normalized transmission becomes weak. In this case, the normalized transmission lies around 1.0. It exceeds 100% transmission at $P_d/(P_{hv}+P_d) = 1$. This means that the loss

of OH in ambient air during the transit in the CMR is smaller than expected from measurements in clean synthetic air. The condition $P_d/(P_{hv}+P_d) = 1$ implies that the total atmospheric OH production continues when the air enters the CMR. Inside the CMR, OH wall reactions compete with gas-phase reactions ($k_{OH}$). When $k_{OH}$ is much larger than $k_w$, the expected steady-state of OH in the CMR is the same value as in outside air, which means that the transmission $\beta_{N2}$ becomes effectively 1.0, and $\beta_{N2}/\beta_{N2}^0 = 1/\beta_{N2}^0$.

## 4.2. Atmospheric OH radical measurements with the FZJ-LIF-CMR during the JULIAC campaign

The JULIAC campaign was designed to investigate tropospheric chemistry in a rural environment which is influenced by biogenic and anthropogenic emissions. The one-year campaign made use of the SAPHIR infrastructure and instrumentation (Section 2.4). During the four intensive JULIAC periods (Table 3 and S3) OH measurements were performed by LIF with chemical modulation. In addition, OH was measured by DOAS for some of the JULIAC periods (winter (I), summer (III), and

autumn (IV) seasons). Besides studying atmospheric chemical processes, the campaign afforded the opportunity to test the chemical modulation technique under natural atmospheric conditions. The CMR was partly operated with different types of injectors and at different propane concentrations (Table S1). For a short time between 1 to 11 February 2019, OH detection was done without the CMR to increase the OH detection sensitivity for measurement of low wintertime OH concentrations.

### 4.2.1 Influence of ambient conditions

The possible influence of ambient conditions on the chemical modulation results was investigated for the conditions of the JULIAC campaign. The ratio $P_d/(P_{hv}+P_d)$ of the dark-to-total OH production was calculated from Reaction R4 ($P_d$) and the photolysis of $O_3$ and HONO ($P_{hv}$). The ratio was between 0.7 and 1.0 most of the time during spring (II), summer (III), and autumn (IV) of 2019 (Figure 9). Only in the winter season (I), the range of values was broader (0.4 - 1.0). The minima of the ratios are due to high NOx concentrations in the morning, which suppresses the HO$_2$ concentration and the OH production by

Reaction R4 until NOx has decreased later in the day. During all seasons, measured OH reactivities were relatively low (< 15 s$^{-1}$) (Table 3) with few exceptions in the winter period where $k_{OH}$ reached 30 s$^{-1}$. For ambient conditions of JULIAC, the predicted influence of $k_{OH}$ on the parameters $\alpha$ and $\beta_{N2}$ remains generally small (Figure 10). The influence on the calculated ambient OH signal is mostly less than 5 % during spring (II), summer (III), and autumn (IV) seasons, and not higher than 12

% in the winter season (I) (Figure 10c). For the encountered $P_d/(P_{h\nu}+P_d)$ ratios of 0.7 - 1.0, the normalized transmission $\beta_{N2}/\beta_{N2}^0$ is only weakly dependent on $k_{OH}$ and the variability is less than $\pm 5\%$ in most cases (Figure 10b). When ambient OH concentrations are calculated (Equation 4 and 6), the influences of $k_{OH}$ on $\alpha$ and $\beta_{N2}$ are combined. The bias, when the influence of $k_{OH}$ is neglected, can be described by the ratio

$$\frac{[OH]^*}{[OH]} = \frac{S_{OH}^*}{S_{OH}} \times \frac{\beta_{N2}}{\beta_{N2}^0} \tag{32}$$

shown in Figure 10d. $[OH]^*$ represents the biased OH concentration that is evaluated with $\alpha^0$ and $\beta_{N2}^0$, whereas the correct $[OH]$ takes the $k_{OH}$ dependence of $\alpha$ and $\beta_{N2}$ into account. For $P_d/(P_d+P_{h\nu})$ ratios between 0.7 and 1.0, the biases from $\alpha$ and $\beta_{N2}$ partly compensate each other. For $P_d/(P_d+P_{h\nu})$ ratios smaller than 0.7, the biases add up and lead to an increasing underestimation of the ambient OH concentration when $k_{OH}$ becomes larger.

Although during the winter season (I) the influence on the calculated ambient OH signal could reach 12%, on average, the combined influence on OH concentration is 2 % for the JULIAC data set. If the corresponding corrections were applied, the amount of useful OH radical data would be reduced by 20 % because not all data (e.g. $HO_2$, $NO$, HONO) needed for corrections were always available. For this reason, no additional corrections were applied to the OH radical JULIAC data set.

### 4.2.2 OH measurement comparison

The combined OH data set measured during the JULIAC campaign by DOAS and LIF-CMR was used for an OH measurement comparison in ambient air. Only summertime (August) data are used, because OH concentrations for the other JULIAC periods were either close to the limit of detection of the instruments (winter and autumn) or the DOAS instrument was not operated (spring). In general, the measurements by LIF and DOAS agree well within their combined $1\sigma$ accuracies of 18% and 6.5 %, respectively (Figure 11). An exception is the heat wave period between 22 and 26 August, when the LIF measurements were

systematically higher than the DOAS measurements by about 25%. The reason for the discrepancy is not clear as no issue with either instrument was found. A linear regression analysis, which considers the precision of both instruments, is done using the algorithm Fitexy (Press, 1992). The scatter plot of the OH data and the regression line forced through the origin are shown in Figure 12. The slope of the regression line is $1.11 \pm 0.02$, where the deviation from unity can be explained by the instrumental accuracies, and therefore no significant systematic error from the calibration with the CMR were found. To see the statistical

error, the residuals of the OH concentrations by LIF obtained from the difference between the measured OH concentration and the linear fit are shown in Figure S2. The residuals scatter symmetrically around zero without curvature, which implies that the linear fit is suitable to describe the relationship between the two OH measurements. Similar good agreement was found in previous OH intercomparisons between the DOAS and FZJ-LIF, when the LIF instrument used wavelength modulation only and data were corrected for the known ozone interference. These intercomparisons were performed in photochemical

degradation experiments in SAPHIR for a wide range of VOCs and chemical conditions (Schlosser et al., 2007; Schlosser et al., 2009; Fuchs et al., 2012a; Novelli et al., 2018; Rolletter et al., 2019; Novelli et al., 2020), and in a field campaign in relatively clean air in North East Germany (Hofzumahaus et al., 1998).

### 4.2.3 OH interferences

The time series of the OH interference measured by chemical modulation during summer (III) season is presented in Figure

11. The interference shows diurnal variations with minimum values at night and maximum values around midday. The median diurnal variation of the interference for the days from 5 Aug. to 2 Sep. has a maximum daytime value of $0.9 \times 10^6$ cm$^{-3}$ and a minimum nighttime value of $0.4 \times 10^6$ cm$^{-3}$. The highest value of $2 \times 10^6$ cm$^{-3}$ occurred in a heat wave from 22 – 29 August, when the daytime air temperature and ozone reached 40°C and 100 ppbv, respectively. At the same time, the total OH

reactivity reached the highest values (19 s⁻¹) with a contribution of reactivity from organic compounds $k_{VOC}$ of up to 14 s⁻¹. During the other seasons, interfering signals at daytime were lower with maximum values of $1 \times 10^6$ cm⁻³ (Figure S3 - S5). Figure 13 compares the OH interferences measured during all four periods of JULIAC with the expected interference from the laser photolysis of ozone. The expected ozone interference is calculated by Equation 19 using the average result from the three experiments in Figure 5. Please note that there is a slight curvature in the parametrization (Figure 13) due to water quenching effect of the OH radical in the detection cell (Holland et al., 2003) by the large range of the water vapour mixing ratio during the JULIAC. Over the whole range of atmospheric conditions, the average measured interferences can be explained by the known ozone water interference within the uncertainty of the parametrization. Over the whole range of atmospheric conditions, the average measured interferences can be explained by the known ozone water interference within the uncertainty of the parametrization. The maximum difference between measured and known interferences is $(3.4 \pm 2.5) \times 10^5$ cm⁻³ for the highest ozone and water vapour concentrations. This difference is well below the limit of detection of the LIF-CMR technique $(7 \times 10^5$ cm⁻³). Thus, there is no evidence for a significant, unexplained OH interference for the atmospheric conditions during the JULIAC campaign (Table 3). Note that the contribution of NO₃ to the interfering signal was negligible for the entire duration of the campaign due to the low NO₃ concentrations that were maximum 10 pptv. Although it is possible that the chemical conditions experienced during the JULIAC campaign were not ideal to observe large interferences as reported in other studies  (Mao et al., 2012; Hens et al., 2014; Novelli et al., 2014a; Feiner et al., 2016; Lew et al., 2020), the good agreement between the CMR-LIF-FZJ and the DOAS indicates that the CMR is correctly implemented. It provides interference-free OH radical concentrations by separating the atmospheric OH signal from the OH radical produced within the detection cell.

## 5 Discussion

The experimental tests in synthetic (Section 3) and in ambient air during the JULIAC campaign (Section 4) demonstrate that the new chemical modulation system of the FZJ-LIF instrument is suitable for measurements of interference-free ambient OH concentrations and internal OH interferences. The theoretically expected dependence of the OH scavenging efficiency and OH transmission on ambient conditions could not be tested. The effects during the conditions of the JULIAC campaign were too small to be clearly detected within the precision of the measurements. Thus, for atmospheric OH reactivities below 15 s⁻¹, it was sufficient to use the CMR parameters that were characterized in synthetic air.

In other field campaigns with higher OH reactivities (> 30 s⁻¹), for example in strongly polluted urban air or in forest environments, the accurate evaluation of OH measurement with the CMR could be more challenging. As shown in Figure 7a and 8a, the residual factor α could significantly increase at high OH reactivities. If not properly corrected, this leads to an overestimation of the calculated interference (Figure 7d). A systematic underestimation of ambient OH signals on the order of (15 - 25) % would be possible (Figure 7c). The sensitivity to $k_{OH}$ can be alleviated by applying higher scavenger reactivities. Some other research groups used much higher scavenger concentrations for their chemical modulation systems (Table 1) with reactivities that are a factor of 5 to 100 higher than in our work (Mao et al., 2012; Novelli et al., 2014a; Rickly and Stevens, 2018; Woodward-Massey et al., 2020). For scavenger reactivities above 10 000 s⁻¹, we expect that the influence of atmospheric $k_{OH}$ on the scavenging efficiency becomes completely negligible. However, different from other groups who did not observe internal OH scavenging in their instruments, we find indications of internal scavenging by 5 % when the injected propane concentration is increased by only a factor 3.8 (Figure 3d). To avoid internal scavenging, which would underestimate the instrumental interference, we kept the scavenger concentration low. Only one other LIF group reported detectable internal scavenging of approximately (5 -10) %, but for a scavenger reactivity of 30 000 s⁻¹ (Woodward-Massey et al., 2020). Currently, it is not clear why the FZJ instrument seems to be more sensitive to internal scavenging than other instruments.

A more serious challenge is the variability of the OH transmission with ambient conditions when no scavenger is added. The transmission ($\beta_{N2}$) needs to be known in order to calculate the ambient OH concentration from the retrieved interference-free ambient OH signal (Equation 6). The transmission in the mode without scavenger (N$_2$ mode) depends on the wall loss rate, the atmospheric OH reactivity, and the transit time through the chemical modulation reactor. The possible influence of atmospheric $k_{OH}$ on the transmission of the FZJ instrument can be seen in Figure 7b. While the impact for the JULIAC conditions is relatively small (Section 4.2.1), a considerable impact would be expected, for example, in a pristine environment like in the Amazonian rain forest. Here, OH reactivities up to 70 s$^{-1}$ were reported (Sinha et al., 2008), while the dark OH production from the reaction of HO$_2$ with NO can be small compared to the total OH production rate. Novelli et al. (2020) estimated the chemical regeneration of OH by Reaction R4 to be 10% in an isoprene emitting forest at 10 pptv NO, which according to the steady-state condition (Equation 20) would correspond to a $P_d/(P_{hv}+P_d)$ ratio of 0.1. For such conditions, the kinetic model for the CMR predicts a decrease of the OH transmission by a factor of 2.5 - 3 at $k_{OH} = 70$ s$^{-1}$. A large correction is required to account for this effect. Novelli et al. (2020) also point out that additional OH regeneration occurs by isomerization of isoprene peroxy radicals, which increases the non-photolytic regeneration rate to about 50 % of the OH loss rate. With a $P_d/(P_{hv}+P_d)$ ratio of 0.5, the predicted decrease of the CMR transmission is still a factor of 1.5. For this particular environment, the chemical mechanism for OH regeneration including processes like RO$_2$ isomerization needs to be known to calculate the CMR transmission.

One possibility to reduce the sensitivity of the transmission with respect to $k_{OH}$ is to shorten the transit time in the CMR. The instrument by the MPI group (Novelli et al., 2014a) uses a residence time which is a factor of 3 - 4 smaller than in our system. A similar time reduction in the FZJ system would lead to much smaller variations of $\beta_{N2}$ (< 10%) at high $k_{OH}$ values. The main disadvantage is the requirement to draw a large air flow in the order of 100 slpm through the CMR. Such a high flow rate has several problems. The current radical source provides a maximum 30 slpm of calibration gas and would need to be newly designed. Second, delivering high flows of clean (synthetic) air for calibration and characterization is expensive. Third, the application in SAPHIR would not be possible because this high flow rate exceeds the total amount of air that can be taken from the chamber during typical operation.

Another solution to quantify the impact of ambient conditions on the OH transmission is to measure the transmission in-situ in the field by taking measurements of OH with and without the CMR in the same air. This directly provides the transmission for the prevailing conditions, but works only reasonably, if OH measurements are interference free. With interference from internal OH, the transmission is overestimated. In order to track changes of the CMR transmission with ambient $k_{OH}$ and diurnal variation of $P_d/P_{hv}$, the transmission needs to be frequently determined. This complicates the automatic measuring operation. A number of groups determined the CMR transmission in the field by this approach, but did not discuss the possible influence of ambient conditions on their results (Table 1). Mao et al. (2012) reported negligible OH loss in field tests, and Woodward-Massey et al. (2020) found less than 5 % sensitivity reduction in ambient air. Both systems used transit times similar to our system. Contrary, the MPI instrument reported lower transmissions of 73 % during daytime for a much shorter residence time. Because of the sparse information about the characteristics of chemical modulation reactors in literature, it is not possible to perform more quantitative comparisons.

Different CMR prototypes built in Jülich were tested with the Peking University LIF instrument (PKU-LIF) in three field campaigns in China (Tan et al., 2017; Tan et al., 2018; Tan et al., 2019). The reactor diameter and total air flow were the same as in this work, but different reactor lengths, injector types, scavenger concentrations and injection flow rates were applied (Table 1). The tests were performed under different ambient conditions in summer 2014 in Wangdu south of Beijing (Tan et al., 2017), in autumn 2014 in Heshan in the Pearl River Delta in southern China (Tan et al., 2019), and in winter 2016 in Huairou north of Beijing (Tan et al., 2018). In each case, the CMR was occasionally mounted for a few hours on the LIF instrument to investigate possible OH interferences. The scavenging efficiency was determined in the field using an OH calibration source. Similar to what is observed during JULIAC, no significant unknown OH interference signals were detected

during the field experiments with the exception of the campaign in Wangdu (Tan et al., 2017) where relatively high level of isoprene (up to 3 ppbv) was observed. Here, unexplained OH signals equivalent to $(0.5 - 1) \times 10^6$ cm$^{-3}$ were found with an experimental 1σ uncertainty of $0.5 \times 10^6$ cm$^{-3}$. Around noontime, these signals were less than 10 % of the ambient OH signals. However, the magnitude of the interference was small compared to the field observations with other LIF instruments

in forested environments (Mao et al., 2012; Hens et al., 2014; Novelli et al., 2014a; Feiner et al., 2016; Lew et al., 2020). The evaluation did not take into account the possible dependence on ambient conditions that is discussed in Section 4. At daytime with OH reactivities of about 15 s$^{-1}$, about half of the unknown interference could have been an artefact due to the unaccounted influence of the atmospheric OH reactivity on the scavenging efficiency, but at nighttime a correction for this influence does not make a difference. Tan et al. (2017) reported also that the sensitivity loss of the OH detection with the CMR was only 5

%. This result was derived from measurements with and without the CMR using an OH calibration source with a longer flow tube (Fuchs et al., 2012a) that does not provide a uniform OH concentration due to its flow profile. CMR transmission measurements with this radical source and with the plug-flow radical source used in this work (Section 3.1) show that the transmission reported by Tan et al. (2017) needs to be corrected from 0.95 to 0.7. The corrected value is in better agreement with the result for the CMR in the present work. The expected small changes do not have any impact on the main findings of

the study, which basically used [OH]$_{WAVE}$ corrected for the known ozone interference.

## 6 Summary and conclusions

A chemical modulation reactor (CMR) for the measurement of interference-free OH radicals with a LIF system was characterized and implemented in the FZJ-LIF instrument. Experiments in the SAPHIR chamber allowed a comprehensive investigation of possible interferences for a wide range of chemical and meteorological conditions.

Several laboratory characterization tests with an OH calibration source enabled an optimization of the operational parameters of the CMR. The reactor has an OH transmission of 64 % at a sample flow of 21 slpm. It was found that a concentration of 19 ppmv of propane by injection with 500 sccm of carrier flow provides efficient mixing and enough OH radical scavenging (96 %) without any significant removal of radicals in the low pressure detection cell (3 %). The total residence time in the reactor is 18 ms and the reaction time for the scavenger is 11 ms.

A comparison with the DOAS instrument in the SAPHIR chamber with humidified synthetic air provided very good agreement to better than 10 % between the two measurements. Interferences from the laser photolysis of ozone in humid air and from NO$_3$ reactions in the detection system known from previous studies with the FZJ-LIF were revisited for the instrument with the CMR in the SAPHIR chamber. The photolytic ozone-water interference is equivalent to an OH concentration of 3.4 $\times 10^5$ cm$^{-3}$ for 50 ppbv O$_3$ and 1% water vapor mixing ratio, in good agreement with what has previously been found

(Holland et al., 2003; Schlosser et al., 2009; Fuchs et al., 2012a). An interference in the presence of NO$_3$ was equivalent to an OH concentration of $0.6 \times 10^5$ cm$^{-3}$ per 10 pptv NO$_3$, a factor of two smaller than what has previously been measured (Fuchs et al., 2016). However, this is not atmospherically relevant.

Four intensive measurement campaigns were performed in four different seasons with sampling ambient air into the SAPHIR chamber (JULIAC). The interferences measured in the summer season had the median diurnal variation of the interference

with a maximum value of $0.9 \times 10^6$ cm$^{-3}$ during daytime and a minimum value of $0.4 \times 10^6$ cm$^{-3}$ at night. The highest interference equivalent to an OH concentration of $2 \times 10^6$ cm$^{-3}$ occurred in a heat wave from 22 – 29 August, when the air temperature and ozone increased to 40°C and 100 ppbv, respectively. This interference could be fully explained by the known ozone interference. No additional unknown interference was observed. Comparison of OH measurements between LIF-CMR and DOAS in ambient air during the JULIAC summer period showed good agreement within measurement uncertainties with

LIF being 13 % larger than DOAS.

A simple kinetic model is presented in this work to estimate the possible influence of ambient chemical conditions on the OH transmission and scavenging efficiency in the CMR. When ambient air is sampled, OH reactions with air pollutants at high OH reactivities begin to compete with OH loss by wall reactions and scavenging. This perturbation depends also on the non-photolytic atmospheric OH production, most importantly the reaction of $HO_2$ with NO, which continues when the sampled ambient air enters the chemical modulation reactor. A correction requires a suite of trace gas measurements (HONO, NO, $HO_2$, $k_{OH}$, and photolysis frequencies). It is found that, for the chemical conditions investigated in this study, the impact on the evaluated OH concentration was small (2 %). In other environments, in particular those with high VOC reactivities and low NOx, OH loss reactions with atmospheric constituents can produce serious perturbations on the transmission and the scavenging efficiency of the CMR unless very high flow rates are used like in the instrument of MPI Mainz. For residence times on the order of 20 ms, as used here and in some instruments of other groups, ambient OH can be depleted in the CMR by more than a factor of 2 by atmospheric reactants. Such perturbations need to be investigated for the specific chemical conditions to evaluate if corrections need to be applied. This means, that chemical modulation, which was developed to eliminate interferences in ambient OH measurements, itself can be subject to interferences that depend on ambient atmospheric conditions. In addition, the OH detection sensitivity, precision as well as time resolution are worse compared to the system without the CMR. So far, for experiments in the SAPHIR chamber, OH radical measurement by the FZJ-LIF has shown good agreement with measurements by DOAS without the need of chemical modulation (Schlosser et al., 2007; Schlosser et al., 2009; Fuchs et al., 2012a; Novelli et al., 2018; Rolletter et al., 2019; Novelli et al., 2020). However, it is not possible to exclude the presence of an unknown interference for all the possible environments. The continued application of chemical modulation will therefore remain an important tool for the detection of OH by LIF.

**Appendix**

**Table A1.** Specification of CMR parameters used in this study.

| Symbol | Definition | Equation number | Value | Unit |
|---|---|---|---|---|
| $S_{ON}$ | On-resonance signal from OH fluorescence and laser-excited background | Eq. 1 | a | Counts mW$^{-1}$ s$^{-1}$ |
| $S_{OFF}$ | Off-resonance signal from laser-excited background | Eq. 1 | a | Counts mW$^{-1}$ s$^{-1}$ |
| $S_{WAVE}$ | OH signal obtained from wavelength modulation | Eq. 1 | a | Counts mW$^{-1}$ s$^{-1}$ |
| $S_{OH}$ | Ambient OH signal obtained from chemical modulation | Eq. 2 and 4 | a | Counts mW$^{-1}$ s$^{-1}$ |
| $S_{OH}^{*}$ | Ambient OH signal obtained from chemical modulation when the influence of $k_{OH}$ on the chemical modulation is neglected. | Eq. 29 | a | Counts mW$^{-1}$ s$^{-1}$ |
| $S_i$ | Interference signal obtained from the chemical modulation | Eq. 2 and 5 | a | Counts mW$^{-1}$ s$^{-1}$ |
| $S_i^{*}$ | Interference signal obtained from chemical modulation when the influence of $k_{OH}$ on the chemical modulation is neglected. | Eq. 30 | a | Counts mW$^{-1}$ s$^{-1}$ |
| $S_{OH}^{N2}$ | OH fluorescence signal obtained from the $N_2$ mode | Eq. 2 | a | Counts mW$^{-1}$ s$^{-1}$ |
| $S_{OH}^{SC}$ | OH fluorescence signal obtained from the scavenger mode | Eq. 3 | a | Counts mW$^{-1}$ s$^{-1}$ |
| $[OH]_{WAVE}$ | OH concentration measured by wavelength modulation | Eq. 1 | a | cm$^{-3}$ |
| $[OH]_{CHEM}$ | OH concentration measured by chemical modulation | Eq. 6 | a | cm$^{-3}$ |
| $[OH]_i$ | OH interference measured by chemical modulation | Eq. 7 | a | cm$^{-3}$ |
| $[OH]_a$ | Steady state ambient OH concentration | Eq.20 | a | cm$^{-3}$ |
| $[OH]'$ | Steady state OH concentration in the CMR | Eq. 21 | a | cm$^{-3}$ |
| $C_{OH}$ | OH detection sensitivity obtained from calibration | Eq. 1 | a | cm$^3$ Counts$^{-1}$ s |

| Symbol | Description | Equation | Value | Units |
|---|---|---|---|---|
| $\alpha$ | Residual factor representing the fraction of ambient OH that is not scavenged by propane in the reactor | Eq.15 | a | b |
| $\alpha^0$ | The residual factor when clean synthetic air is carrier gas for OH | Eq. 16 | 0.042 | b |
| $\beta$ | General function to calculate the OH transmission of a tube section of the CMR | Eq.24 | a | b |
| $\beta_{tube}^0$ | OH transmission of the CMR tube without built-in injectors when clean synthetic air is carrier gas for OH | Eq.8 | 0.81 | b |
| $\beta_{N2}$ | OH transmission of the CMR for the $N_2$ mode | Eq.6 | a | b |
| $\beta_{N2}^0$ | OH transmission of the CMR for the $N_2$ mode when clean synthetic air is used as carrier gas for OH | Eq.8 | 0.64 | b |
| $\beta^e$ | OH transmission in the entrance section | Eq.26 | a | b |
| $\beta^{e,0}$ | OH transmission in the entrance section when clean synthetic air is carrier gas for OH | Eq.10 | 0.92 | b |
| $\beta_{N2}^r$ | OH transmission in the reaction section ($N_2$ mode) | Eq.27 | a | b |
| $\beta_{N2}^{r,0}$ | OH transmission in the reaction section (N2 mode) when clean synthetic air is carrier gas for OH | Eq.12 | 0.69 | b |
| $\beta_{SC}^r$ | OH transmission in the reaction reactor section (scavenger mode) | Eq.28 | a | b |
| $\beta_{SC}^{r,0}$ | OH transmission in the reaction section (scavenger mode) when clean synthetic air is carrier gas for OH | Eq.13 | 0.03 | b |
| f | Fraction of the total CMR flow which does not contain scavenger | Eq. 17 | 0.04 | b |
| $k_w$ | OH wall loss rate coefficient in the CMR tube without built-in injectors | Eq.9 | 11.8 | $s^{-1}$ |
| $k_w^e$ | OH wall loss rate coefficient in the entrance section of the CMR tube | Eq.10 | 12.3 | $s^{-1}$ |
| $k_w^r$ | OH wall loss rate coefficient in the reactor section of the CMR tube | Eq.12 | 33 | $s^{-1}$ |
| $k_{SC}$ | Scavenging rate coefficient of OH by propane in complete homogenous mixing case | Eq.14 | 513 | $s^{-1}$ |
| $k_{SC}^{eff}$ | Effective scavenging rate coefficient of OH by propane obtained from scavenging efficiency tests | | 283 | $s^{-1}$ |
| $k_{SC}'$ | Scavenging rate coefficient of OH by propane in the fraction containing no propane in case of inhomogeneous mixing | | 0 | $s^{-1}$ |
| $k_{SC}''$ | Scavenging rate coefficient of OH by propane in the fraction containing propane in case of inhomogeneous mixing | Eq.17 | 534 | $s^{-1}$ |
| $\Delta t$ | The transit time through the CMR tube | Eq.8 | 17.8 | ms |
| $\Delta t_r$ | The transit time through the reactor section | Eq.16 | 11.2 | ms |

| $\Delta t_e$ | The transit time through the entrance section | Eq.9 | 6.6 | ms |

a Variable

b Dimensionless

## Code and data availability

## Author contributions

AH designed JULIAC campaign and organized it together with HF and FH. CC performed the laboratory experiments, analysed the data, and wrote the paper together with AN and AH. All co-authors contributed with data and to the discussion of the manuscript.

## Competing interests

The authors declare that they have no conflict of interest.

## Acknowledgements

This project has received funding from the European Research Council (ERC) under the European Union's Horizon 2020 research and innovation program (SARLEP grant agreement no. 681529). We thank to Birger Bohn for the provision of photolysis frequencies for JULIAC campaign and Zhaofeng Tan for the discussion about the CMR conditions during the field campaigns in China.

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

**Tables and figures**

Table 1. Characteristics of chemical modulation reactors for different OH LIF instruments.

| Parameters | FZJ [a] | PKU [b] | MPIC [c] | PSU [d] | IU [e] | UL [f] |
|---|---|---|---|---|---|---|
| Volume flow (slpm) | 21 | 21 | 150-280 | 7 | 3-9 | 32 |
| Residence time (ms) [g] | 18 | 22 / 18 [h] | not specified | 100 | not specified | 20 |
| Reaction time (ms) [i] | 11 | 19 / 11 [h] | 2.5-4 | 25 / 10 [j] | 50 | 20 |
| Scavenger | $C_3H_8$ | $C_3H_8$ | $C_3H_8$ | $C_3F_6$ | $C_3F_6$ | $C_3H_8$ |
| Scavenger concentration (ppmv) [k] | 19 | 3 - 500 | 100 | 150 - 500 | 400 - 1200 | 110 or 1100 |
| Scavenger reactivity [k, l] $(s^{-1})$ | 540 | 70 - 13500 | 2700 | 7700 - 26000 | 20000 - 60000 | 3000 or 30000 |
| OH scavenging (%) [m] | 96 [n] | 80 - 97 [n] | 80 - 95 [o] | 94 [o] | > 90 | > 99 [n] |
| OH transmission (%) [p] | 64 [n] | 70 [n] | 73 [q] | ~ 100 [j] | ~ 100 | > 95 [o, q] |
| Reference | This study | (Tan et al., 2017; Tan et al., 2018) | Novelli et al. (2014a) | Mao et al. (2012) | Rickly and Stevens (2018) | Woodward-Massey et al. (2020) |

a Forschungszentrum Jülich, Germany.

b Peking University, Beijing, China.

c Max-Planck-Institute for Chemistry, Mainz, Germany.

d Pennsylvania State University, PA, USA.

e Indiana University, IN, USA.

f University of Leeds, UK.

g Total transit time.

H The shorter time applys to the CMR version as used in Tan et al. (2018).

i Transit time from the injectors to the inlet.

j The shorter time was used by Feiner et al. (2016).

k Assuming homogeneous mixing in the CMR flow.

l The scavenging reactivity was calculated assuming homogeneous mixing. The used rate coefficients for $C_3H_8$ and $C_3F_6$ + OH radical are $1.1 \times 10^{-12}$ and $2.08 \times 10^{-12}$ cm$^3$ s$^{-1}$ at 298K, respectively. (Dubey et al., 1996; Atkinson et al., 2006)

m Corresponds to $(1- \alpha) \times 100\%$.

n Measured in synthetic air.

o Measured in ambient air with OH produced photolytically by Hg lamp radiation.

P Corresponds to $\beta_{N2} \times 100\%$.

q Measured in ambient air with natural OH at daytime.

**Table 2.** Specification of instruments used at SAPHIR in this study.

| Species | Measurement technique | Time resolution / s | Limit of detection (1σ) | Accuracy (1σ) |
|---|---|---|---|---|
| OH | LIF-CMR | 270 | $0.7 \times 10^6$ cm$^{-3}$ | 18 % |
| OH | DOAS | 134 | $0.8 \times 10^6$ cm$^{-3}$ | 6.5 % |
| HO$_2$ | LIF | 47 | $0.8 \times 10^7$ cm$^{-3}$ | 18 % |
| OH reactivity ($k_{OH}$) | LP-LIF | 180 | 0.3 s$^{-1}$ | 18 % |
| Photolysis frequencies | Spectroradiometry | 60 | [a] | 10 % |
| O$_3$ | UV photometry | 40 | 1 ppbv | 5 % |
| NO$_X$ (NO+NO$_2$) | Chemiluminescence | 180 | NO: 4 pptv NO$_2$: 2 pptv | 5 % |
| HONO | LOPAP | 180 | 5 pptv | 10 % |
| NO$_3$ | CRDS | 1 | 3 pptv | 30 % |
| H$_2$O | CRDS | 60 | 0.1 % [b] | 5 % |

a Several orders of magnitude lower than the maximum value at noon.

b Volume mixing ratio

**Table 3.** Summary of meteorological conditions and trace gas concentrations during JULIAC given as daytime median values with $1\sigma$ standard deviations of ambient variabilities.

| | OH / $10^6$ cm$^{-3}$ | NO / ppbv | $O_3$ / ppbv | $H_2O$ / %[a] | $k_{OH}$ / s$^{-1}$ | $k_{VOC}$[b] / s$^{-1}$ | T / ˚C | $j(O^1D)$ / $10^{-6}$ s$^{-1}$ | $P_d$ / ppbv s$^{-1}$ | $P_{hv}$ / ppbv s$^{-1}$ |
|---|---|---|---|---|---|---|---|---|---|---|
| | 0.27 | 0.3 | 22.3 | 0.6 | 6.1 | 2.2 | 4.7 | 0.3 | 0.2 | 0.15 |
| 14 Jan. – 11 Feb. | (±0.8) | (±6.2) | (±9.7) | (±0.2) | (±6.0) | (±2.2) | (±4.0) | (±0.4) | (±0.2) | (±0.1) |
| | 1.6 | 0.2 | 41.6 | 0.7 | 5.8 | 3.0 | 15.6 | 2.0 | 1.9 | 0.6 |
| 9 Apr. – 6 May | (±2.3) | (±1.7) | (±17.5) | (±0.2) | (±2.6) | (±1.4) | (±7.5) | (±2.7) | (±0.8) | (±0.2) |
| | 3.0 | 0.2 | 38.5 | 1.4 | 6.3 | 3.2 | 26.6 | 3.6 | 1.9 | 0.4 |
| 4 Aug. – 2 Sep. | (±3.2) | (±0.5) | (±18.0) | (±0.3) | (±2.9) | (±2.3) | (±6.9) | (±3.7) | (±1.0) | (±0.2) |
| | 0.4 | 0.8 | 16.8 | 0.8 | 5.6 | 1.7 | 8.4 | 0.5 | 0.9 | 0.1 |
| 28 Oct. – 24 Nov. | (±1.1) | (±2.8) | (±11.1) | (±0.2) | (±3.5) | (±1.5) | (±4.5) | (±0.7) | (±0.5) | (±0.1) |

a Volume mixing ratio

b OH reactivity of non-methane VOCs, calculated as the difference between measured total $k_{OH}$ and the sum of calculated reactivities of $CH_4$, CO, $O_3$, NO, and $NO_2$.

**Table 4.** Transmission and wall-loss rate coefficients for OH and $HO_2$ in the chemical modulation reactor (1/8" injectors) when nitrogen is injected.

| | OH | | $HO_2$ | |
|---|---|---|---|---|
| | $k_w$ / $s^{-1}$ | β | $k_w$ / $s^{-1}$ | β |
| Entrance section[a] | 12.3 | 0.92 | 5.5 | 0.96 |
| Reaction section[b] | 33 | 0.69 | 14.5 | 0.85 |
| Total CMR | - | 0.64 | - | 0.82 |

a Transit time is 6.6 ms.

b Transit time is 11.2 ms.

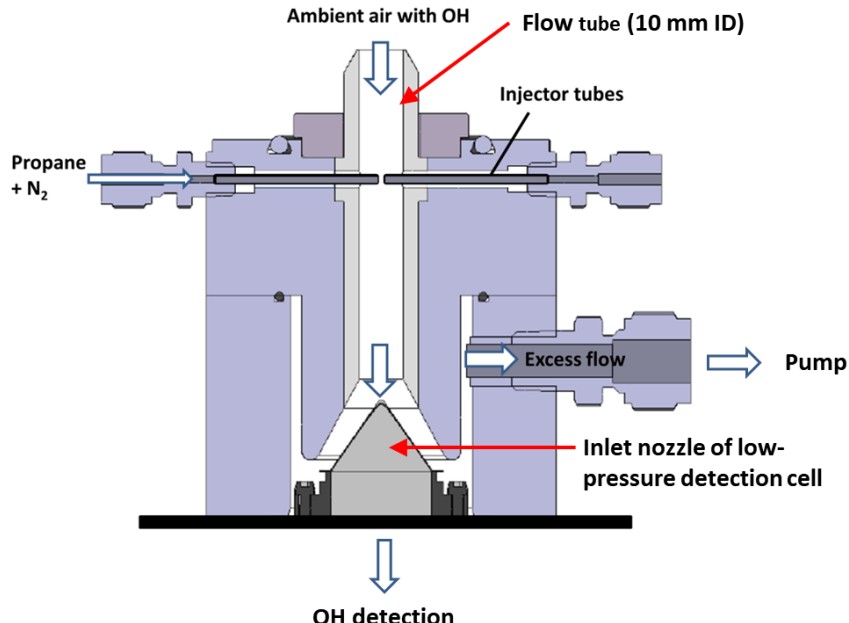

**Figure 1.** Schematic drawing of the chemical modulation reactor (CMR) in front of the OH LIF detection cell. Ambient air is drawn through a PFA flow tube (79 mm length; 10 mm inner diameter). A flow of 1.1 slpm is expanded through an 0.4 mm inlet nozzle into a low-pressure cell for OH detection and 20 slpm of excess air are removed by a pump. A nitrogen flow of 0.5 slpm containing 19 ppmv propane as a OH scavenger is added via two 1/8" (OD) injector tubes (50 μm ID) 50 mm above the nozzle. The distance between the injector tips is less than 1 mm. All flows (nitrogen flow, excess flow, nozzle flow) are held constant during operation.

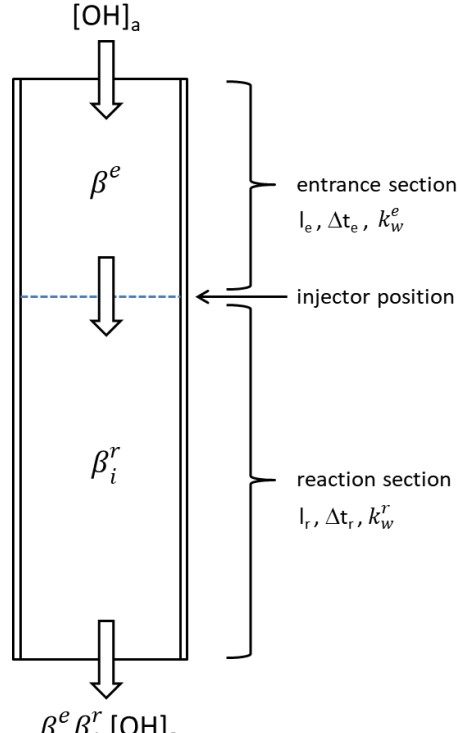

**Figure 2.** Schematic of the chemical modulation reactor (CMR). The flow tube has an entrance section with a length $l_e$ up to the injector position. In this section, OH radicals are lost by wall reaction during the transit time $\Delta t_e$ with a rate coefficient $\mathbf{k_w^e}$, resulting in an OH transmission $\boldsymbol{\beta^e}$. Behind the injector position follows the reaction zone, where OH is scavenged when propane is injected into the flow. The reaction zone has a length $l_r$ with a transit time $\Delta t_r$ and wall loss rate coefficient $\mathbf{k_w^r}$. The OH transmission $\boldsymbol{\beta_i^r}$ in the reaction section depends on the chemical modulation mode ($i = N_2$ when pure nitrogen is injected, $i = sc$ when a scavenger-nitrogen mixture is injected). The total transmission for OH is given by the product $\boldsymbol{\beta^e \beta_i^r}$.

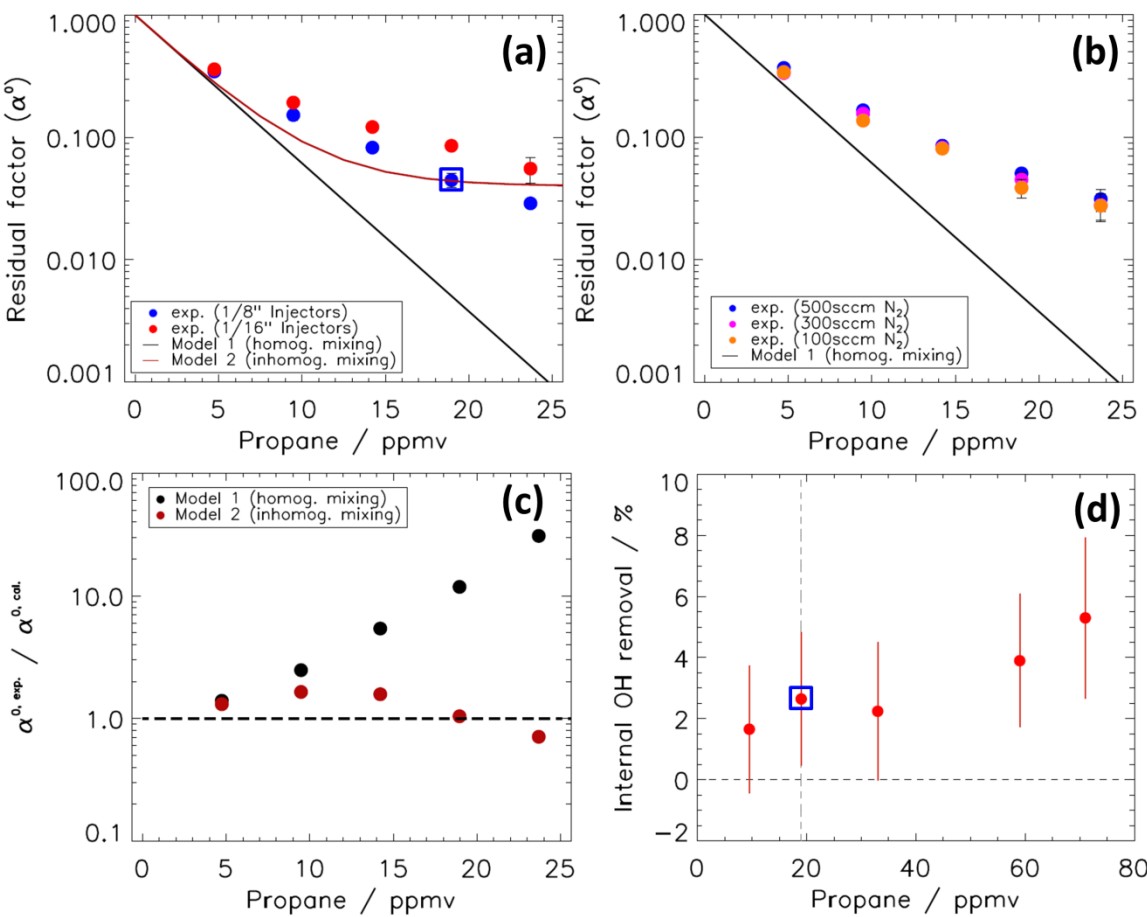

**Figure 3. a.** Residual factor ($\alpha^0$) for OH in a flow of humidified synthetic air running through the CMR. The OH air mixture is provided by a photolytic OH radical source. $\alpha^0$ is shown as a function of the calculated concentration of propane (at 298 K and 1 atm), which is injected as a scavenger with 500 sccm $N_2$ into the CMR flow. The experimental $\alpha^0$ values are shown for two types of injectors with different outer diameter (1/8" and 1/16"). The blue open squares indicate the condition used for normal operation in this work. Measurements are compared to calculations assuming either homogeneous or inhomogeneous mixing. **b.** Same as **a.**, showing measured $\alpha^0$ values for different injector flow rates. **c.** Ratio of experimental and modelled $\alpha^0$ for the case with 1/8" injectors and 500 sccm $N_2$ injector flow. **d.** Measured OH removal by internal scavenging inside the OH detection cell (see text).

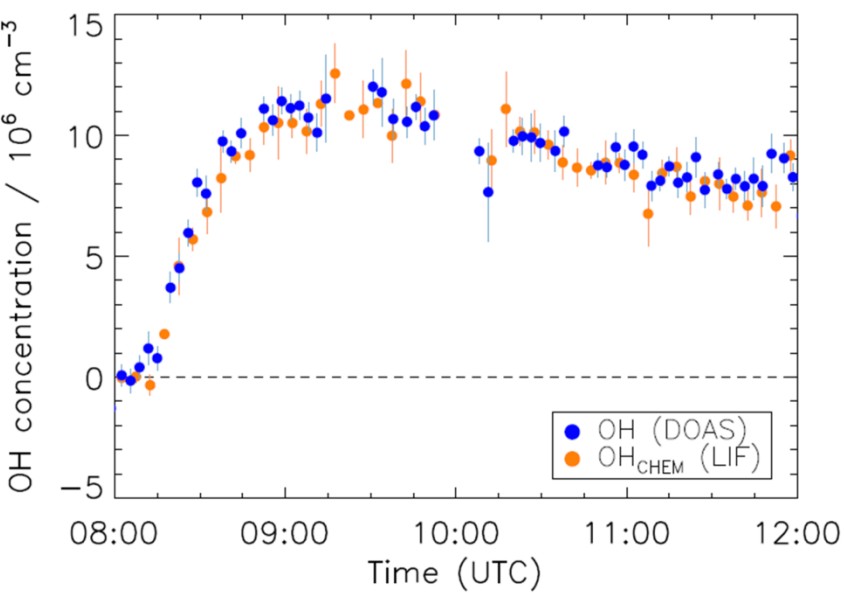

**Figure 4.** Time series of OH concentrations measured by DOAS and LIF (with CMR) in humidified synthetic air in the SAPHIR chamber. The measured OH reactivity in the chamber air was less than 2 s$^{-1}$. The chamber was illuminated by solar radiation and OH was mainly produced by solar photolysis of traces of nitrous acid emitted from the chamber walls. The data shown are averaged over 300 s.

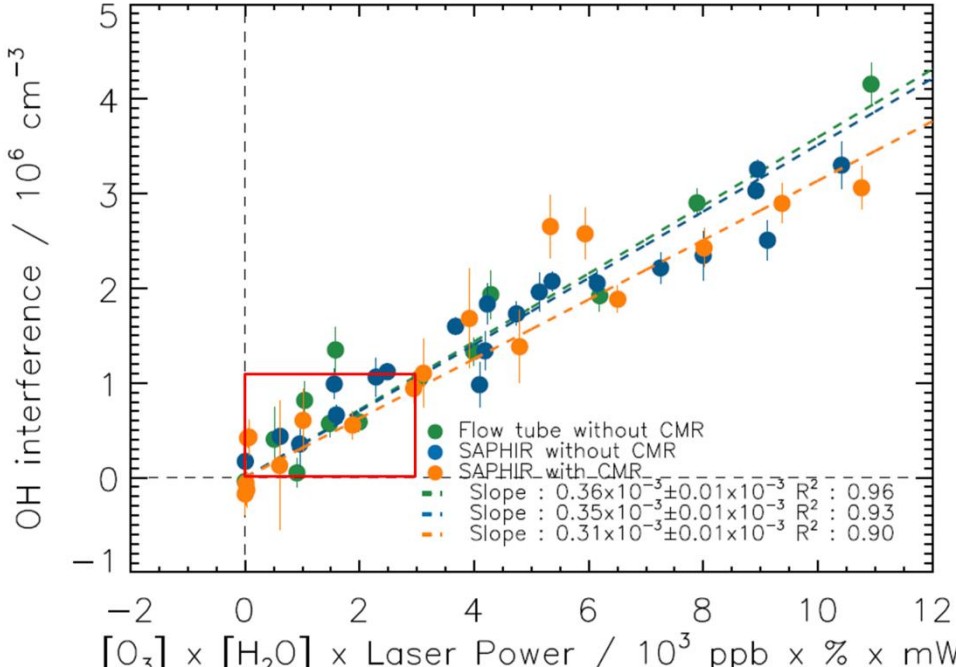

**Figure 5.** OH interference, expressed as equivalent ambient OH concentration, from the laser photolysis of ozone in humidified synthetic air for a range of concentrations of ozone, water vapour, and laser power. Tests were done with and without the CMR with a flow tube or in chamber experiments. Measurements were fitted to a linear function forced through the origin. The red box specifies the interference for
5   concentrations of ozone and water vapour normally observed in the lower troposphere.

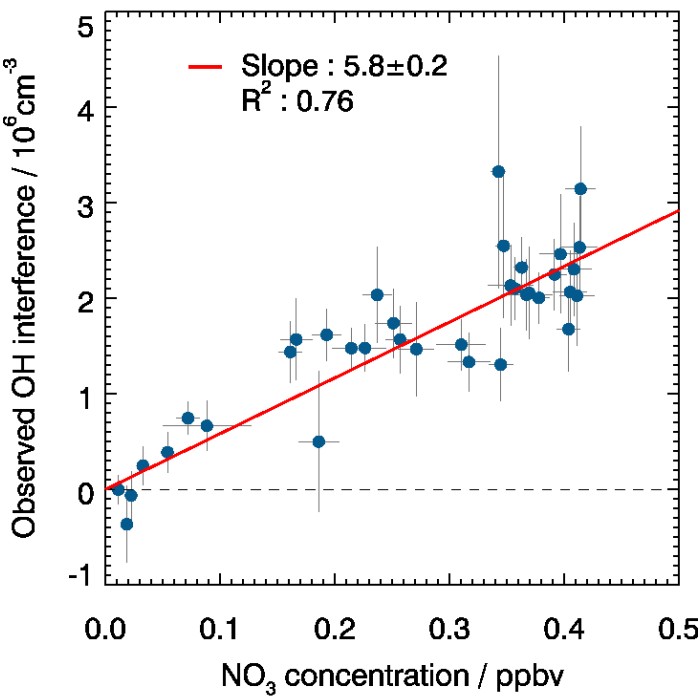

**Figure 6.** Correlation between the measured OH interference, determined by chemical modulation, and the $NO_3$ concentration measured during an experiment when $NO_3$ (from thermal decomposition of frozen $N_2O_5$ crystals) was present in the chamber. Measurements were fitted to a linear function forced through the origin.

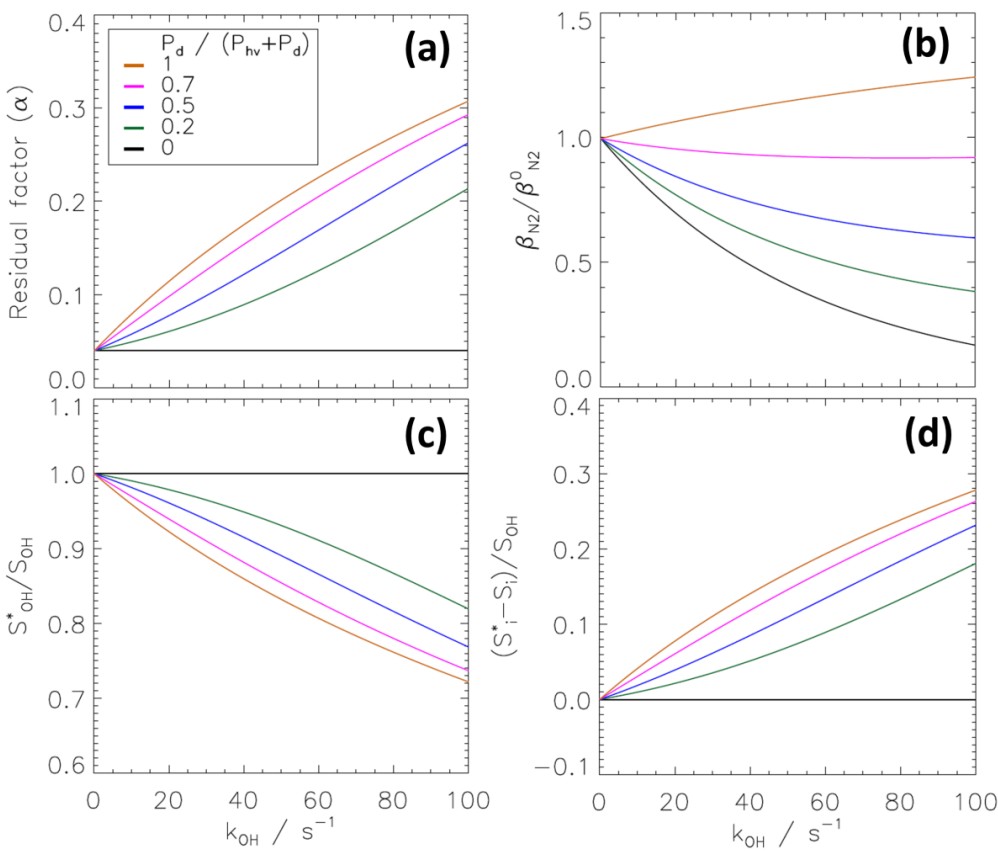

**Figure 7.** Theoretical $k_{OH}$ dependence of the residual factor $\alpha$ (**a.**) and normalized transmission $\beta_{N2}/\beta^0_{N2}$ (**b.**). The applied model (Section 4.2) assumes homogeneous mixing. Parameters are taken from the characterization experiments of the CMR. The dependence is shown for different ratios of dark to total atmospheric OH production rates. A ratio of one corresponds to nighttime (without photolytic OH production), while a ratio of zero denotes a case without dark OH production. Panels c. and d. show the bias in the determination of the ambient OH signal ($S_{OH}$) and interference signal ($S_i$), if the $k_{OH}$ dependence of $\alpha$ is not taken into account. $S_{OH}*$ and $S_i*$ are calculated with $\alpha^0$ instead of $\alpha(k_{OH})$.

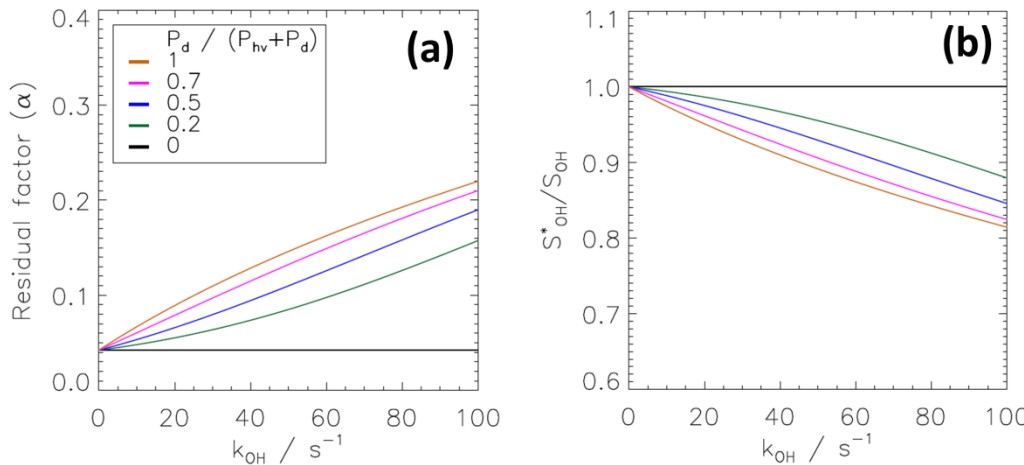

**Figure 8.** Calculated $k_{OH}$ dependence of the residual factor $\alpha$ and ratio $S^*_{OH}/S_{OH}$ using a model that takes inhomogeneous mixing of the scavenger into account (see Section 4.3). The dependence is shown for different ratios of dark to total atmospheric OH production rates. The colored horizontal lines denote the same ratios of dark to total OH production rates as used in Figure 7.

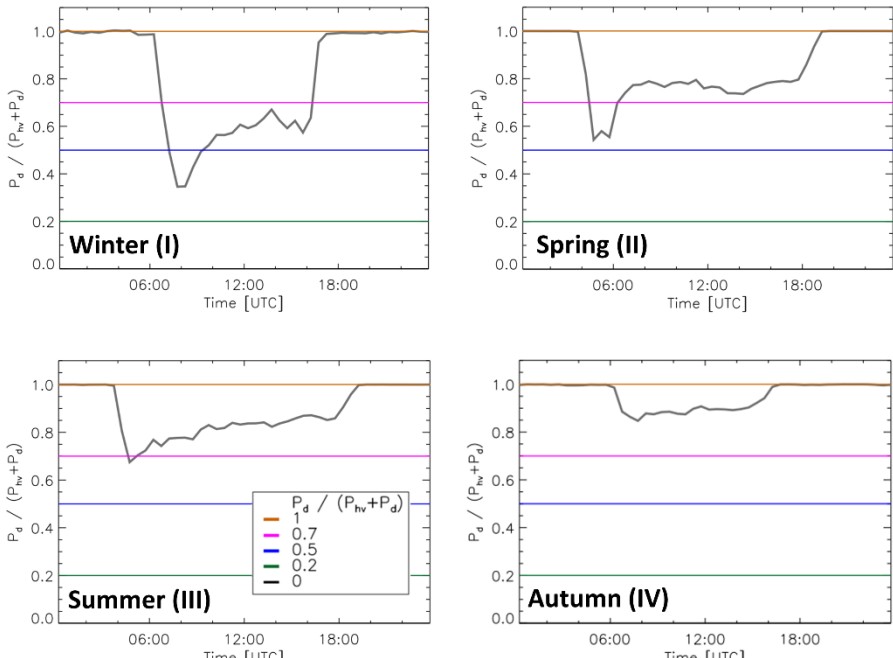

**Figure 9.** The median ratio of dark to total atmospheric OH production rates during four JULIAC periods (Grey lines). $P_d$ is the calculated reaction rate of $HO_2$ with NO (Reaction R4), and $P_{h\nu}$ is the OH production rate from the photolysis of ozone (Reaction R1 + R2) and HONO (Reaction R3). For the calculations, measured quantities (300s averaged) were used (Table 2). The colored horizontal lines denote the same ratios of dark to total OH production rates as used in Figs. 7 and 8.

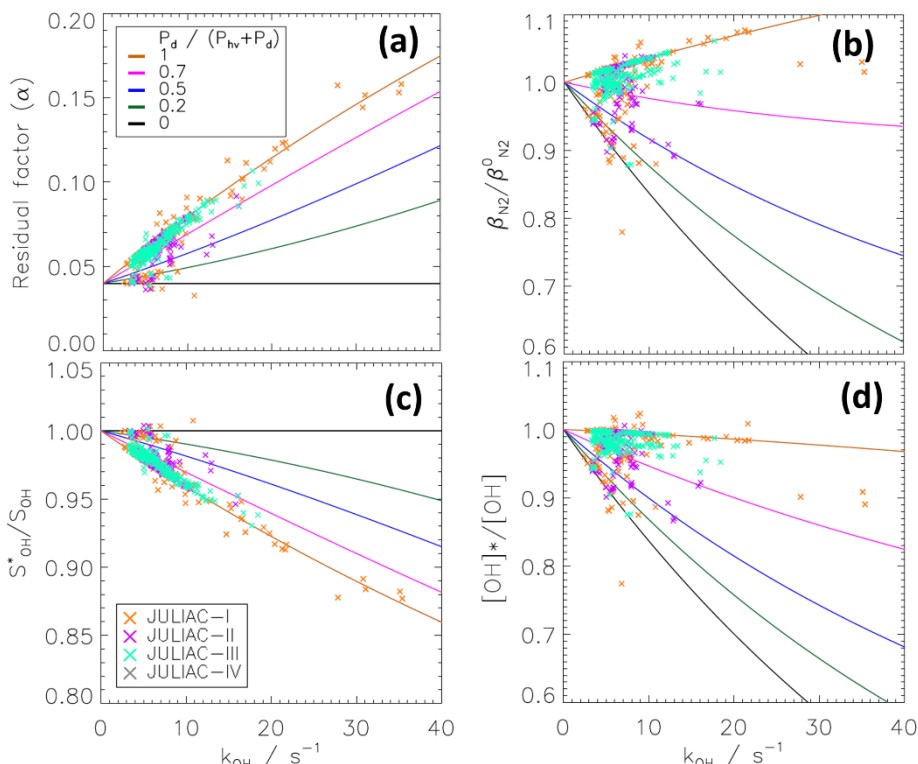

**Figure 10.** Modelled $k_{OH}$ dependence of the CMR properties for the atmospheric chemical conditions during the JULIAC periods in different seasons of 2019. Here, the model for a homogeneously mixed scavenger was applied. See Figure 7 for further explanations of (a) - (c). Panel (d) shows the ratio of corrected to uncorrected ambient OH concentrations, if the $k_{OH}$ dependence of $\alpha$ and $\beta_{N2}$ is taken into account.

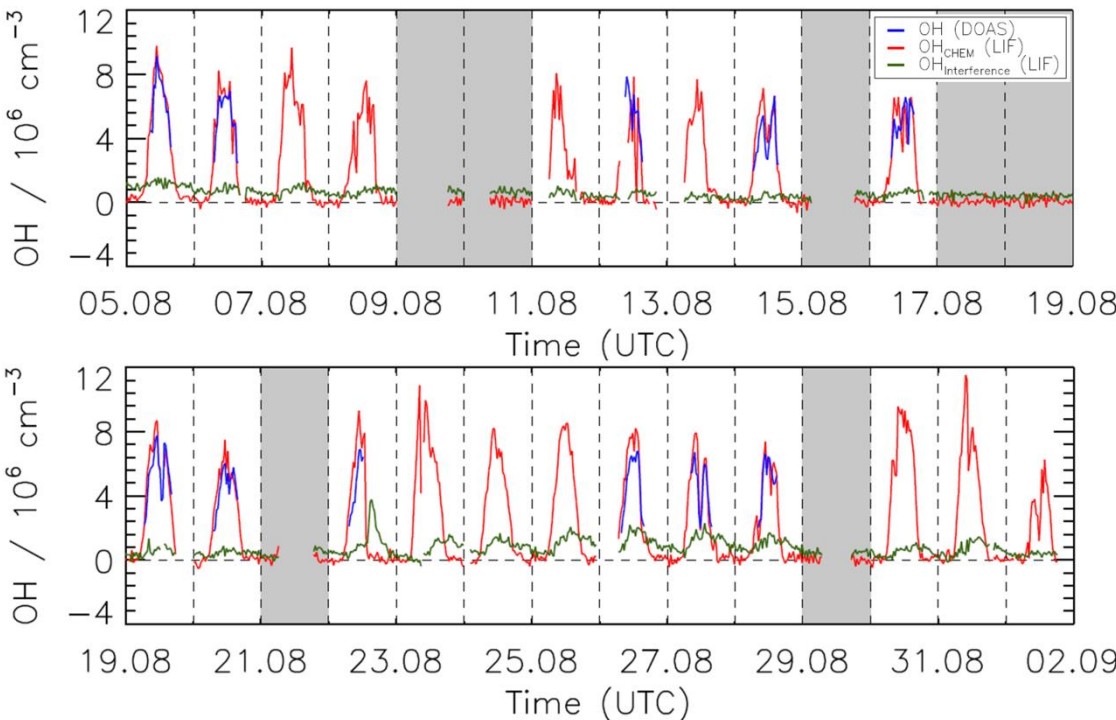

**Figure 11.** Time series of OH concentrations measured by LIF-CMR (red line) and DOAS (blue line) during the summer period of the JULIAC campaign. The green line shows the equivalent OH concentration of the interference signal in the LIF system. All data points are 30 min averages. Vertical dashed lines denote midnight. Grey shaded areas indicate when the chamber roof was closed. Data gaps occurred when measurements were stopped for calibrating instruments.

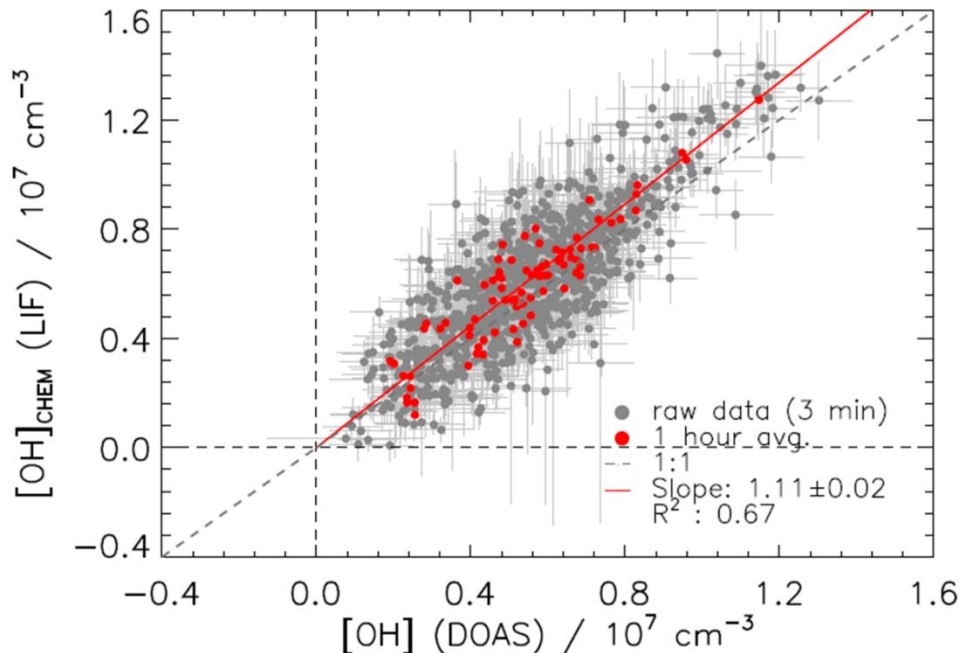

**Figure 12.** Correlation plot between OH concentrations measured by the FZJ-LIF-CMR and the DOAS instruments during the summer period of the JULIAC campaign except for the heat wave period between 22 and 26 August. The 1436 data points are averaged over 3 minute and 1 hour. The red line is a linear fit of 3 minute data set weighted with the statistical errors of both instruments and forced to the origin. Vertical bars denote the 1σ precision of the measured data points.

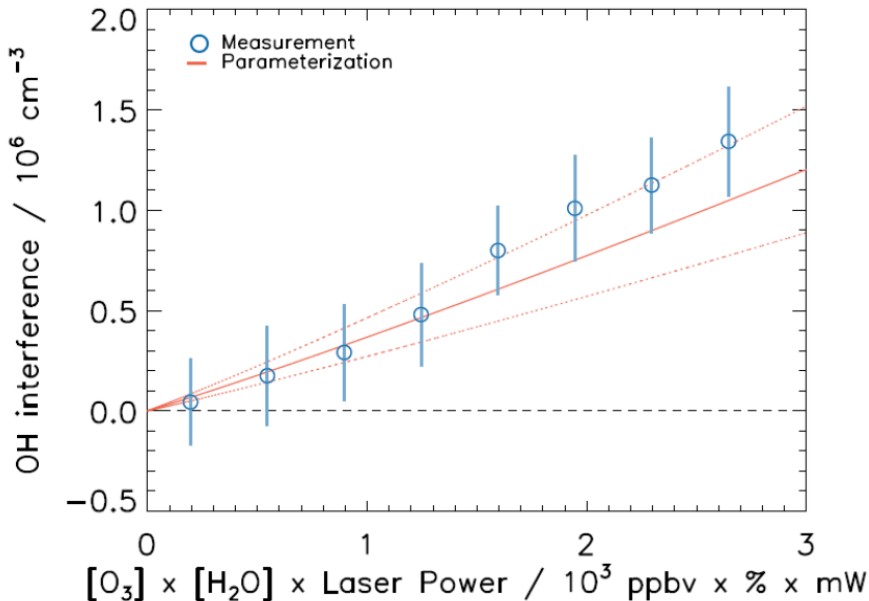

**Figure 13.** Averaged OH interference measured during all JULIAC periods (vertical bars are standard deviations of the bin averaged data) compared to a parameterization (Equation 19, Section 3.5.1) determined in laboratory experiments. The red dotted lines represent the uncertainty of the parameterization. Each bin is averaged by more than 100 data points.