# Peer review of "Characterization of a chemical modulation reactor (CMR) for the measurement of atmospheric concentrations of hydroxyl radicals with a laser-induced fluorescence instrument"

_Atmospheric Measurement Techniques, 2020_

## Referee Comment (RC1) · Anonymous Referee #1 · 7 Oct 2020

This manuscript describes the application and characterization of a chemical modulation reactor for interference-free measurements of OH using the FZJ-LIF instrument. I find it interesting that while applying a method to remove interferences, additional interferences could be introduced. This makes the tests performed and reported in this manuscript very important for consideration in future ambient OH measurements. This manuscript is thorough and well written. With only minor changes, I fully recommend publication in AMTD.

Specific comments

[Figure]

P7, line 2: How far do the injectors protrude into the CMR?

P9, Eq. 17: Clarify what $\alpha2$ is in the text.

P9, line 4: Although the measurements suggest no internal OH scrubbing, the model calculated reactivity (550 sˆ-1) is greater than that calculated for complete mixing (283 sˆ-1$\times$1.8=510 sˆ-1). Do the authors have any comment on this?

P10, line 35: Although the three experiments shown in Figure 5 agree to within 15%, do the authors have any comment on why the interference is lower with the CMR?

P14, lines 21-4: This was confusing to read. Is line 22 supposed to read 'if the ratio Pd/(Ph$\nu$+Pd) is "smaller" than 0.7'?

P40, Figure 13: It appears that the measurements show curvature while the parameterization does not. Do the authors have any comment on this?

Technical corrections

Abstract. P1, line 20: add the word 'for' to be read as 'allowed for performing'

Abstract. P1, line 26: add an 's' to be read as 'ambient air conditions'

Abstract. P1, line 27: change to 'in the summer season had a median'

Abstract. P1, line 32: change to 'for an unexplained'

P2, line 15: change to 'helped with investigation'

P2, line 18: remove 'the' to be read as 'where OH reactivity'

P2, line 19: remove comma after 'was found'

P2, line 20: change 'The discrepancy' to 'This discrepancy'

P2, line 21: replace 'in' to be read as 'missing from the chemical mechanisms'

P4, line 26: add 'an' to be read as 'used as an OH'

P4, line 27: remove comma after scavenger

P4, line 31: remove second 'is' to be read as 'It is controlled'

P5, line 18: change 'an $1\sigma$' to 'a $1\sigma$'

P5, line 20: remove 's' from 'measurements'

P5, line 20: add a comma after '(HOx cell)'

P5, line 22: reword to be read as 'During the JULIAC experiments, NO ($2.5\times10^{13}$ cm$^{-3}$) was used for minimizing a possible'

P6, line 1: add a space after '270'

P6, line 2: change 'higher' to 'greater'

P6, line 5: remove comma after '%)'

P6, line 9: remove comma after 'SAPHIR'

P6, line 10: add 'a' to read as 'by a mixed'

P6, line 21: remove 'Here,'

P7, line 1: add space after 'Reynolds'

P7, line 4: remove comma after 'found'

P7, line 23: remove comma after 'OH'

P7, line 24: remove comma after 'expected'

P7, line 27: remove comma before 'if'

P8, line 16: reword to be read as 'injectors produce a larger flow resistance within the CMR'

P11, line 15: add 'the' to be read as 'for the FZJ-LIF'

P12, line 20: add an 's' to 'Equation' for both instances

P13, line 18: add an 's' to 'kind'

P13, line 34: add 'the' to be read as 'in the case'

P14, line 8: add 'a' to be read as 'is a similar'

P14, line 21: change 'relative' to 'relatively'

P14, line 22: remove comma after 'error'

P15, line 7: add comma after 'seasons'

P15, line 12: replace 'depending' with 'dependent'

P15, line 15: remove comma after '%'

P16, line 3: remove 'from organic compounds'

P16, line 24: replace 'in' with 'on'

P16, line 28: add a space to read '10 000'

P17, line 9: add a comma after 'environment'

P17, line 15: add 'a' to be read as 'provides a maximum'

P17, line 21: replace 'reasonable' with 'reasonably'

P18, line 26: add 'the' to be read as 'in the summer'

P18, line 28: correct spelling to 'occurred'

P18, line 38: add a comma after 'study'

P18, line 41: replace 'in' to 'on' to be read as 'times on the order'

P26, line 1: replace 'gases' with 'gas'

P28, line 3: replace 'an' with 'a'

P40, line 3: add 'to' to be read as 'compared to a'

P40, line 4: add '.' after 'experiments'

P40, line 5: add 'by' to be read as 'averaged by more'

Table S1 caption: add 'the' to be read as 'for the JULIAC'

Table S2 caption: replace 'gases' with 'gas'

Figure S2 caption: remove 'and DOAS'

---

## Referee Comment (RC2) · Anonymous Referee #2 · 13 Nov 2020

**Review of "Characterization of a chemical modulation reactor (CMR) for the measurement of atmospheric concentrations of hydroxyl radicals with a laser-induced fluorescence instrument" by Cho et al.**

This paper presents information on the use of a pre-reactor designed to minimize interferences in the measurements of hydroxyl radicals (OH) by the technique of laser-induced fluorescence (LIF). The use of such reactors has become common-place since the recognition of interferences in LIF OH measurements that are not accounted for by spectral modulation (on-line/off-line). This so-called chemical background has been found to be systematically and sometimes significantly larger than the spectroscopic background.

This paper points out that there is known and correctable interference from the production of OH within the instrument from the photolysis of ozone by the laser in the presence of water vapor. In principle, this interference will be captured by the chemical method of background determination.

Important in the use of an external reactor, when working with highly reactive radical species such as OH, is the impact of the reactor surfaces on ambient OH and whether the effect of the surfaces that are contacted by the sample changes between the signal plus background and the background measurement modes. Also, it is important to evaluate the impacts of gases added to ambient air on the performance of the LIF detection.

These issues were covered quite well in this paper and are presented in an understandable fashion.

**General comments.**

The CMR was presented as a design with a configuration that was considered optimum and final. Did the researchers consider other configurations of the CMR, including (1) reduced pressure at say 100-200 mbar to optimize mixing, (2) switching the addition of propane between the front location and near the end of the reaction zone to provide constant composition entering the LIF detection cell (like the OH CIMS technique), (3) test various diameters and lengths of reaction, and various materials for the reactor walls, and (4) examine impacts of flow rate on mixing and CMR performance? Such tests, if performed, should be discussed in the paper. If not performed, their possible impacts should be presented

While some instruments have apparently not seen significant interferences beyond the impact of ozone photolysis by the probe laser, is it possible that it is because of the atmospheric environments in which the tests were done. Indeed, even in the present study, there doesn't appear to be a significant impact of biogenic VOCs on OH reactivity. Previous studies have indicated that forested environments, presumable with significant BVOC emission, have the largest measured interferences. This possibility should be added to discussion, including in the middle of page 3.

This reviewer found the use the various Greek symbols ($\alpha$, $\beta$) with multiple superscripts and subscripts not clearly defined. While Figure 2 was helpful in this regard, not all of the symbols are included. It might be easier for the reader if the symbols were more mnemonic. In other words, use T for transmission of radicals (as on page 12) throughout the paper with a clear definition (the fraction of radicals successfully transported through the CMR), and another symbol (perhaps R) for the reverse of transmission, which is referred to as the residual factor (suggest a different term) and definition (the fraction of radicals lost during transport through the CMR). This way it is easier for the reader to remember what the various symbols stand for. This reviewer suggests the authors consider a table with the symbols used in the paper to aid the reader.

This reviewer tried to reproduce the plots in Figure 7 based on information in the paper. It was not possible arrive at the exactly the same figures. Suggest that you carefully check the calculations (perhaps by at least two people independently) to ensure that these calculations are correct.

**Specific comments.**

Page 1, lines 11 & 12. Suggest replacing the word "how" used twice in this first sentence. For example, "…are essential to investigate mechanisms for oxidation and transformation of trace gases…" and "…and processes leading to the formation of secondary pollutants…"

Page 1, line14. The use of chemical modulation has been utilized in most LIF instruments "recently". Somewhere, if possible, give a reference to the first suggestion, or a recent suggestion, that this is how such measurements should be performed.

Page 1, line 17. While this reviewer agrees that this paper describes the application and characterization of the CMR, it is not obvious that it has been validated. Suggest adding text, perhaps near the end of the Introduction that demonstrates the CMR has been validated. Later, on page 3, the words "introduced, described, and characterized" are used, which might be better here as well.

Page 1, line 20. Suggest rewording "It allowed performing OH…". Perhaps "enabled" instead of "allowed" or a reorganization of the sentence.

Page 1, line 22. Suggest removing "A" to read "Good agreement was obtained…".

Page 1, line 22. Suggest a "-" or "/" or "vs" in "LIF-DOAS".

Page 1, lines 22-23. This reviewer takes issue that good agreement between LIF and DOAS confirms that the CMR provides interference-free OH measurements. It depends on whether the comparison was performed in conditions that cause interference, and it assumes that there is no possibility of interference in DOAS. While the latter seems unlikely, it cannot be ruled out completely. In fact, the statement on line 30, indicates that the situation of JULIAC did not include unexplained interference. Thus, it might not be suitable to use the setup described in a forested environment with large BVOC emissions. This should be discussed somewhere in the paper, and perhaps the case made not quite to strong in the abstract.

Page 1, lines 27-28. The description of the medium diurnal variation of the interference could be misleading. I think you mean that the maximum value of $0.8 \times 10^6$ occurs during daytime, and the minimum value occurs at night, but it could be interpreted other ways. Suggest rewording the sentence.

Page 1, lines 33-37. The models described in the paper do not seem to be chemical kinetic models (nor kinetic chemical models). Also, the language could be improved in these sentences. For example, "A kinetical chemical model of the chemical modulation reactor was developed…". Suggest substituting a different word to avoid using "chemical" twice or rearranging the sentence. In the next sentence, "reactions" and "reactor' are used close together that makes the sentence a bit awkward. This reviewer prefers "photolytic" to "photolytical", but which one the author uses is their choice.

Page 1, lines 40-41. This conclusion that chemical modulation can be subject to interferences while attempting to correct for other interferences is an important finding. It should be clearly stressed (maybe with a bit more detail) here and elsewhere in the paper.

Page 2, line 14. The Stone et al., 2012 reference discussing OH measurements is good, but now a bit out of date. Suggest adding one or two more recent references.

Page 2, lines15-16. Suggest rewording part of this sentence "…which helped in the investigation of the atmospheric OH radical budget."

Page 2, lines 28-29. Suggest rewording "…in atmospheric chemical models increases the predicted concentrations of OH considerably for some conditions, for example…"

Page 2, line 33. It might make sense to add discussion of measurements of OH by other techniques and the relationship between measurements and models. There is indication, for example, that CIMS could be prone to interferences but it is believed that they are accounted for by the way propane is used in the inlet.

Page 2, line 37. Suggest adding one or more reference to "Previous studies" that are mentioned, again including measurements of from CIMS as well as LIF.

Page 3, lines 1-2. The Hofzumahaus and Heard reference does not suggest that use of chemical modulation necessarily leads to OH measurements that are interference free. Suggest rewording this sentence.

Page 3, lines 10-11. Again, while some instruments see large interferences, it may because of where they were deployed rather than the configuration of the instrument (or a combination). Suggest rewording and/or adding text to make this clear.

Page 3, line 8. Again, the term "interference-free" is used. This has not been unequivocally demonstrated. Indeed, the FZJ use of the CMR leads to a reduction of the interference, but not complete removal as shown later in the paper.

Page 3, lines 16-17. Suggest rewording "…but nighttime observations were similar to those found in forested environments."

Page 3, line 24. Suggest changing "avoid" to "minimize".

Page 3, line 30. Suggest "…is developed that provides estimates of the possible interferences…".

Page 3, line 36. Suggest "…a large range of chemical…".

Page 4, lines 5-6. Suggest rewording "…the excitation wavelength is modulated on and off the peak of the OH absorption line…".

Page 4, line 8. Could you describe why the timing of on-line and off-line signal measurements are not the same? Normally, optimum signal to noise is achieved when they are the same, unless the noise of the background (off-line) is very small.

Page 4, line 13. Suggest adding text to indicate that the intensity of the mercury lamp radiation must be known, and is determined by actinometry.

Page 4, line 17. Suggest rewording "Hydroxyl radicals originating not from ambient air but formed within the detection cell…".

Page 4, line 19. The physical location of the CMR is said to be "on top". While this may be true, up or down do not really apply to gas flows. Suggest using "in front" instead.

Page 4, line 20 and following. The dimensions and details of the CMR are described, but it doesn't say why these parameters were selected. Suggest adding text describing the method by which the current design was arrived at.

Page 4, line 21. The injectors are 1/8" OD tubes, with 50 µm ID, correct? Suggest rewording.

Page 4, line 25. For CIMS measurements, it was found that the purity of the propane is very important. Even small amounts of impurities can cause problems. Please give the purity of the propane use to make the propane/$N_2$ mixture. Indicate whether experiments with different sources (vendors) of propane have been performed. This is very important.

Page 4, line 30. Why are the measurement cycles so much longer with the CMR in place?

Page 4, line 33. Why do you do on-line and off-line measurements while using the CMR? This should not be necessary in normal measurement mode (but perhaps useful when doing tests). Equations 4 and 5 show that it isn't necessary to measure $S_i$.

Page 5, line 8. Suggest "…the OH cell with the CMR in place…".

Page 5, lines 10-13. This conclusion may not be valid if there are contaminants in the propane.

Page 5, line 17. The DOAS does not require calibration, but it does depend on knowledge of the absorption cross section for the pressure and temperature conditions of the measurement, and the absorption path length. Please add this information to this section.

Page 5, line 22. It is probably better to say "…experiments an NO concentration…", since the reader is likely to read "an en oh" rather than "a nitric oxide".

Page 6, line 1-2. Suggest "…which has a light transmission of greater than 0.8 over the complete solar spectral range…".

Page 6, line 3. Suggest "…to prevent contamination from…".

Page 6, line 6. The beginning of this section says that comprehensive tests were performed, but that is not clear from what is written. Did you do any tests with biogenic VOCs and ozone added to synthetic air to see if other interferences showed up? How about $NO_3$ oxidation experiments of BVOCs? These are situations that could produce additional interferences and should be investigated and discussed.

Page 6, line 9. Suggest "…measurements in SAPHIR during which ambient air was…".

Page 6, line 11. Suggest "…located close to the city of Julich."

Page 6, line 8-25. It is not clear why the experiments were performed in the SAPHIR chamber rather than just by sampling ambient air. Please add a sentence or two of explanation.

Page 6, line 21. Suggest "The total OH reactivity…".

Page 6, line 23. Suggest "range" instead of "suite".

Page 6, line 24. JULIAC indeed allowed potential interferences to be investigated, but only within the range of conditions found in Julich. There are other conditions throughout the world that are different and could present other interferences. This should be stated either here or elsewhere in the paper.

Page 6, line 35. It is not clear why the term "laminar flow tube" is used immediately followed by stating that it exhibits plug flow conditions. While it is possible to experience approximate plug flow in a tube, this is usually found at low pressures (< a few mbar) and in the entrance region (before the full flow regime is established). It is likely with the calibrator to be due to the latter. It would be useful to state the Reynolds number and remind the reader of the flow regime, both in the calibrator and the CMR. Then indicate why the calibrator might exhibit plug flow.

Page 7, line 1. Suggest "…(Reynold number, $R_e$ = 2800)…". This value of $R_e$ is actually in the transition regime (2300-2900), so the flow cannot be clearly classified as turbulent. Also, there is an transition

region of up to tube diameters to establish the condition of a given $R_e$. Finally, the injectors generate turbulent (to enhance mixing), so that has an effect as well.

Page 7, line 7. The experiment to measure transmission without the injectors is useful, but it would be interesting to see how it varies with flow rate and tube diameter. Were such experiments done?

Page 7, line 9. Since the radical calibrator produces both OH and $HO_2$, is there evidence of loss (or do you expect it from the kinetics) from the reaction of OH with $HO_2$. It would be helpful to give the range of OH concentrations employed in the various tests performed, including this one.

Page 7, line 12. Suggest "The rate coefficient, $k_w$, that was obtained can be used…"

Page 7, line 20. Was the concentration of CO in the CMR varied to see whether it had an effect? Are you concerned that if there was surface conversion of $HO_2$ to OH that could be released into the gas-phase, it would be masked by the presence of CO. It seems that the $HO_2$ loss experiment could be conducted without addition of CO, assuming reaction between $HO_2$ and OH was not an issue (comment earlier).

Page 7, line 27. There is a comma at the beginning of the line. Suggest rewording the text before and after equation (11) so this is not needed.

Page 7, line 33. Perhaps remind the reader that the following discussion is for clean air, in other words $k_{OH} = 0$. You also might want to use the superscript 0 to indicate this in all the terms being presented. It is also not clear the superscripts "e" and "r" were defined on the previous page.

Page 8, line 3. Suggest "where $k_{sc}$ is the pseudo first-order rate constant for reaction between OH and propane."

Page 8, line 5. Suggest "The fraction of ambient OH remaining that subsequently enters the detection cell…".

Page 8, line 12. Suggest "…as expected from Equation 16, whose derivation assumes homogeneous mixing…".

Page 8, line 27. This reviewer disagrees that further increases in the propane concentration gives only small improvements in $\alpha$. Looking at the data rather than the red line, there is little indication that the curve is flattening. The y-axis on this plot is logarithmic with magnitude orders of magnitude. Perhaps four symbol diameters correspond to a factor of two change in the residual amount remaining. Because of the issues with too much propane in the detection cell, experiments should be performed with longer length reactor tubes. It may be that a 50% increase is sufficient to bring $\alpha$ close to zero. This reviewer suggests performing such tests before publication of this paper.

Page 8, line 33. Is it also possible that ultraviolet light, either from the radical source or ambient sunlight enters the CMR and affects the conversion chemistry? This possibility should be discussed along with data collected using the apparatus.

Page 9, Line 9. Suggest "As employed in the $HO_2$ transmission tests…"

Page 9, line 5-25. It appears to this reviewer that higher amounts of propane could be used in the CMR with minor impacts in the detection cell. For reasons that are not clear, a 3% effect is considered negligible, where a 5% effect is unacceptable. It is excellent that this range of propane levels was tested to measure this affect. This reviewer suggests such range for the scavenging efficiency tests.

Page 9, line 33. "photolytical" is used again.

Page 9, line 35. Suggest "The instruments agreed to well within the combined 1σ accuracies…". Can you demonstrate this numerically, for example by providing the mean and standard deviation of the differences between the two measurements (or some other metric)?

Page 10, line 1. When regression is mentioned throughout the paper, please indicate whether it is standard least squares or, as mentioned in one case, a bivariate technique such as fixexy in Press.

Page 10, line 8. This was mentioned before, but why is it necessary to do the on- and off-resonance measurements when using the CMR?

Page 10, line 15. Suggest "…when air pollutants influence the chemistry in…"

Page 10, line 15. Indicate the level of uncertainty (1σ?) for the accuracy statement (here and other places in the paper).

Page 10, line 19. Suggest "…known interferences (ozone photolysis and $NO_3$ radicals)…"

Page 10. Line 21. Suggest "…interferences from other sources." This is because there could be sources other than chemical compounds, perhaps.

Page 10, line 28. Suggest "…and water vapour were varied up to 450 ppbv and 1.8%..."

Page 10, line 34. It is stated that the ozone photolysis interference is linear with the product of ozone and water vapor, but the parameterization in Figure 13 is not linear. Please give more information.

Page 10, lines 30-32. Why are the measurement techniques in this experiment different from Table 2?

Page 11, line 8. Suggest "…chamber using the thermal decomposition of $N_2O_5$ added from a condensed source which produces $NO_3$ radicals. Indicate measurement technique for $NO_3$ in these experiments. Can you explain the large amount of scatter in the Figure 6 plot?

Page 11, line 19. Suggest "This section describes studies of the application….".

Page 11, line 33. As mentioned earlier, how do you know that there is no photolysis in the CMR? Suggest not using "inner darkness", but use a more precise term.

Page 12, line 2. Suggest "…is the dominate non-photolytic source…". (This reviewer agrees that photolytic is the correct term here).

Page 12, line 6. It should be recognized that in the normal atmosphere, the ratio of $HO_2$ to OH can be quite large, perhaps 100 or much more. In that case, a conversion of a small fraction of $HO_2$ to OH could greatly change the amount. Suggest reworking the discussion to reflect this.

Page 12, line 8. It is not true that NO is determined primarily NO reaction with ozone. This could be true, but there are also situations when the reaction of NO with peroxy radicals ($HO_2$ and $RO_2$) is comparable or even larger than $NO+O_3$. Suggest rewording this sentence.

Page 12. As a reader, I was confused by this section of equations. It was not clear how the terms and symbols used earlier applied to this section. This is why this reviewer suggests a table or figure to present all the various symbols and terms used. It is also not clear why there are two terms for transmission, β and T. With the correct corresponding subscripts and superscripts, one should be enough.

Page 13, line 1. It should be noted that $k_{OH}$ is never 0. It can be as low as 0.5 $s^{-1}$ for very clean air, but all air has methane and CO.

Page 13, line 5. Suggest reminding the reader why the minimum value is 0.042. Also, after using $\beta$ values up to this point, why are the plots of $\alpha$ values?

Page 13, first paragraph. Mention that Figure 7b will be discussed later, in Section xx.

Page 13, line 8. Suggest "…the production $P_d$ exceeds the photolysis-based OH production…"

Page 13, line 13. Suggest removing comma "…would be obtained if the influence…"

Page 13, line 15. Suggest "…signal $S_i*$ is similarly :"

Page 13, line 18. Instead of "all kinds of" give the range of values over which it applies. It could be for the full range of values studied.

Page 13, line 28. Suggest "The result presented in Figure 8a is very similar to the homogeneous mixing case shown in Figure 7a."

Page 13 line 36 and page 14, line 1. What is meant by "uncertainty of the model". Suggest being clearer here or remove this assertion. It seems that real differences are expected between the two models.

Page 14, line 6. It is not clear why the inverse of the term in equation 31 is plotted in Figure 7b.

Page 14, line 12-13. This is a very important point that needs to be clearly stated in the abstract and the summary.

Page 14, line 17. Suggest "…OH production continues when the air enters…".

Page 14, line 18. Suggest "…When $k_{OH}$ is much larger than…".

Page 14, line 21. Suggest "…causes a relatively small…".

Page 14, line 29. Suggest "…measured by DOAS for some of the JULIAC periods…".

Page 14, lines 30-31. Suggest "…the campaign afforded the opportunity to test the chemical…".

Page 14, line 32-33. Suggest "For only a short time between 1 to 11 February 2019, OH detection was done…".

Page 15, line 12. Suggest "…is only weakly dependent on $k_{OH}$ and…".

Page 15, line 14. What does "…and otherwise add up" mean? This needs some more explanation.

Page 15, line 14. It seems that the average correction is not so important, but rather the size and frequency of larger corrections. If the corrections are always small, there is no need to use the CMR. If they are are sometimes big, and you don't know when, then it is important to have the CMR capability. Suggest rewording this part

Page 15, line 19. Is this better for "The combined OH summer dataset measured…"?

Page 15, line23. Can the authors provide some metric to indicate that the two LIF measurements indeed agree within their combined uncertainties?

Page 15, line 23. It should be noted that the differences between the two OH measurements were during the heatwave.

Page 15, line 24. Suggest "… systematically higher than the DOAS measurements by about 25%."

Page 15, line 26. See earlier comment about regression and fixexy.

Page 16, line 6. Is it meant to refer to equation 19 rather than equation 12?

Page 16, line 11. Has the limit of detection versus averaging time been explored? It would be good to calculate the Allan-Werle variance for the LIF-CMR instrument and discuss in this paper. The data in Figure 13 have hours of data included, so the uncertainties could potentially be much smaller.

Figure 1. Some suggestions to the body of the paper also apply to the figure caption. Suggest "Schematic drawing" rather than Technical drawing. Suggest "in front" rather than "on top". Suggest providing the mixing ratio rather than "traces of propane".

Figure 2. Possibly include other symbols in this figure (see suggestions made earlier).

Figure 4. Suggest "The data shown are…".

Figure 5. Suggest using the same x=axis on this figure and Figure 13.

Figure 7. In the body of the paper, it would be good to describe why these various terms are being plotted. What are each of these factors meant to show to the reader?

Figure 9. The values in the plots do not seem to agree with the measurement period averages in Table 3. For example, for winter compared to summer, the value of $j(O^1D)$ is about 10 times larger in summer, while NO is about 3 times higher in winter. This seems to indicate that the fraction of OH from photolysis should be about three times higher in summer than winter. But the figure seems to indicate than the photolysis fraction is about twice as high in the winter as the summer. Suggest checking the calculations for this figure and Table 3.

Figure 13. Suggest giving the equation number for the parameterization. Adjust text as appropriate based on earlier comment.

Table S1. Note that the detection limit depends on the averaging time (see earlier comment), so provide averaging time corresponding to the detection limit shown.

Figures S1, S2 and S3. Suggest "Dashed lines denote midnight."

---

## Author Comment (AC1) · 1 Jan 2021

*Response to the reviewers for the* **"Characterization of a chemical modulation reactor (CMR) for the measurement of atmospheric concentrations of hydroxyl radicals with a laser-induced fluorescence instrument" study** *by* **Changmin Cho et al.**

We would like to thank the reviewers for careful reading the paper and providing useful comments which improved the manuscript. We reply here to each comment in blue color.

**Anonymous Referee #1**

**Comment #1:** P7, line 2: How far do the injectors protrude into the CMR?

**Response:** The injectors are inserted approximately 4 mm into the flow tube. We changed the text P7, line 2 to be 'which protrude approximately 4mm into the flow tube'.

**Comment #2:** P9, Eq. 17: Clarify what α2 is in the text.

**Response:** We changed the text in P9, line 2 to be '…with $\alpha_2$ $(k_{sc}^{''} = k_{OH+propane}[Propane])$…'

**Comment #3:** P9, line 4: Although the measurements suggest no internal OH scrubbing, the model calculated reactivity (550 sˆ-1) is greater than that calculated for complete mixing (283 sˆ-1×1.8=510 sˆ-1). Do the authors have any comment on this?

**Response:** The value of 550 s$^{-1}$ given for the model calculated reactivity for the inhomogeneous mixing case ($k_{sc}^{''}$ , Section 3.2) was erroneous. The correct value is 534 s$^{-1}$ and it is obtained from the calculated reactivity for the complete mixing case, 513 s$^{-1}$ ($k_{sc}$, Section 3.2), divided by 0.96. It is larger than $k_{sc}$ as the higher concentration of propane is assumed to be present in a smaller volume (96% of the total volume). We have clarified this in the text at the end of section 3.2.

**Comment #4:** P10, line 35: Although the three experiments shown in Figure 5 agree to within 15%, do the authors have any comment on why the interference is lower with the CMR?

**Response:** The reviewer raises a good point and we also tried and understand if there was any connection between a lower interference being observed when the CMR is used. With the CMR, we measure the $O_3$ concentration in the excess air of the CMR. However, without the CMR, the O3 concentration was monitored in the excess air of the radical source. This could maybe constitute part of the difference observed. In addition, the two experiments without the CMR were done in 2016, while the experiment with CMR was executed in 2019 and small changes on the LIF detection cell (e.g., different nozzle and therefore different flow pattern in the cell) happening between the tests can affect the interference. We believe that the difference (15%) of the interference between with and without the CMR and over a span of 3 years represents our limit in reproducibility and for this reason we included this value in figure 13. It is important to mention that the observed interference agrees well (within 30 %) with previous values (Holland et al., 1998; Holland et al., 2003) as shown in the section 3.5.1.

**Comment #5:** P14, lines 21-4: This was confusing to read. Is line 22 supposed to read 'if the ratio Pd/(Phν+Pd) is "smaller" than 0.7'?

**Response:** We have removed the last paragraph of section 4.1.4 as it was redundant and did not have additional information.

**Comment #6:** P40, Figure 13: It appears that the measurements show curvature while the parameterization does not. Do the authors have any comment on this?

**Response:** The interference signal showed in figure 13 represent the total interference measured with the CMR. This can be fully explained by the known $O_3/H_2O$ interference within the uncertainty of the parametrization from laboratory and chamber tests with synthetic air. It is still possible that other interferences such as OH from the dissociation of stabilized Criegee intermediates produced from ozonolysis of alkenes, from photolysis of acetone, from $NO_3$ radicals could contribute to the total interference up to $3.4 \times 10^5$ cm$^{-3}$ for higher ozone (80 ppbv) and water vapour (1.6 %) concentrations observed together with high ambient temperature (up to 40 ℃). However, these differences are below the limit of detection of FZJ-LIF-CMR ($7 \times 10^5$ cm$^{-3}$).

In addition, the parameterization shows a slight curvature due to water quenching (see the respond to the review #2 comment in Page 10, line 34)

Technical corrections

**Comment #7:** Abstract. P1, line 20: add the word 'for' to be read as 'allowed for performing'

**Response:** Done.

**Comment #8:** Abstract. P1, line 26: add an 's' to be read as 'ambient air conditions'

**Response:** Done.

**Comment #9:** Abstract. P1, line 27: change to 'in the summer season had a median'

**Response:** Done.

**Comment #10:** Abstract. P1, line 32: change to 'for an unexplained'

**Response:** Done.

**Comment #11:** P2, line 15: change to 'helped with investigation'

**Response:** Done.

**Comment #12:** P2, line 18: remove 'the' to be read as 'where OH reactivity'

**Response:** Done.

**Comment #13:** P2, line 19: remove comma after 'was found'

**Response:** Done.

**Comment #14:** P2, line 20: change 'The discrepancy' to 'This discrepancy'

**Response:** Done.

**Comment #15:** P2, line 21: replace 'in' to be read as 'missing from the chemical mechanisms'

**Response:** Done.

**Comment #16:** P4, line 26: add 'an' to be read as 'used as an OH'

**Response:** Done.

**Comment #17:** P4, line 27: remove comma after scavenger

**Response:** Done.

**Comment #18:** P4, line 31: remove second 'is' to be read as 'It is controlled'

**Response:** Done.

**Comment #19:** P5, line 18: change 'an 1σ' to 'a 1σ'

**Response:** Done.

**Comment #20:** P5, line 20: remove 's' from 'measurements'

**Response:** Done.

**Comment #21:** P5, line 20: add a comma after '(HOx cell)'

**Response:** Done.

**Comment #22:** P5, line 22: reword to be read as 'During the JULIAC experiments, NO ($2.5 \times 10^{13}$ cm^-3) was used for minimizing a possible'

**Response:** Done.

**Comment #23:** P6, line 1: add a space after '270'

**Response:** Done.

**Comment #24:** P6, line 2: change 'higher' to 'greater'

**Response:** Done.

**Comment #25:** P6, line 5: remove comma after '%)'

**Response:** Done.

**Comment #26:** P6, line 9: remove comma after 'SAPHIR'

**Response:** Done.

**Comment #27:** P6, line 10: add 'a' to read as 'by a mixed'

**Response:** Done.

**Comment #28:** P6, line 21: remove 'Here,'

**Response:** Done.

**Comment #29:** P7, line 1: add space after 'Reynolds'

**Response:** Done.

**Comment #30:** P7, line 4: remove comma after 'found'

**Response:** Done.

**Comment #31:** P7, line 23: remove comma after 'OH'

**Response:** Done.

**Comment #32:** P7, line 24: remove comma after 'expected'

**Response:** Done.

**Comment #33:** P7, line 27: remove comma before 'if'

**Response:** Done.

**Comment #34:** P8, line 16: reword to be read as 'injectors produce a larger flow resistance within the CMR'

**Response:** Done.

**Comment #35:** P11, line 15: add 'the' to be read as 'for the FZJ-LIF'

**Response:** Done.

**Comment #36:** P12, line 20: add an 's' to 'Equation' for both instances

**Response:** Done.

**Comment #37:** P13, line 18: add an 's' to 'kind'

**Response:** Done.

**Comment #38:** P13, line 34: add 'the' to be read as 'in the case'

**Response:** Done.

**Comment #39:** P14, line 8: add 'a' to be read as 'is a similar'

**Response:** Done.

**Comment #40:** P14, line 21: change 'relative' to 'relatively'

**Response:** Done.

**Comment #41:** P14, line 22: remove comma after 'error'

**Response:** Done.

**Comment #42:** P15, line 7: add comma after 'seasons'

**Response:** Done.

**Comment #43:** P15, line 12: replace 'depending' with 'dependent'

**Response:** Done.

**Comment #44:** P15, line 15: remove comma after '%'

**Response:** Done.

**Comment #45:** P16, line 3: remove 'from organic compounds'

**Response:** Done.

**Comment #46:** P16, line 24: replace 'in' with 'on'

**Response:** Done.

**Comment #47:** P16, line 28: add a space to read '10 000'

**Response:** Done.

**Comment #48:** P17, line 9: add a comma after 'environment'

**Response:** Done.

**Comment #49:** P17, line 15: add 'a' to be read as 'provides a maximum'

**Response:** Done.

**Comment #50:** P17, line 21: replace 'reasonable' with 'reasonably'

**Response:** Done.

**Comment #51:** P18, line 26: add 'the' to be read as 'in the summer'

**Response:** Done.

**Comment #52:** P18, line 28: correct spelling to 'occurred'

**Response:** Done.

**Comment #53:** P18, line 38: add a comma after 'study'

**Response:** Done.

**Comment #54:** P18, line 41: replace 'in' to 'on' to be read as 'times on the order'

**Response:** Done.

**Comment #55:** P26, line 1: replace 'gases' with 'gas'

**Response:** Done.

**Comment #56:** P28, line 3: replace 'an' with 'a'

**Response:** Done.

**Comment #57:** P40, line 3: add 'to' to be read as 'compared to a'

**Response:** Done.

**Comment #58:** P40, line 4: add '.' after 'experiments'

**Response:** Done.

**Comment #59:** P40, line 5: add 'by' to be read as 'averaged by more'

**Response:** Done.

**Comment #60:** Table S1 caption: add 'the' to be read as 'for the JULIAC'

**Response:** Done.

**Comment #61:** Table S2 caption: replace 'gases' with 'gas'

**Response:** Done.

**Comment #62:** Figure S2 caption: remove 'and DOAS'

**Response:** Done.

**Anonymous Referee #2**

**General comments.**

**Comment #63**: The CMR was presented as a design with a configuration that was considered optimum and final. Did the researchers consider other configurations of the CMR, including (1) reduced pressure at say 100 - 200 mbar to optimize mixing, (2) switching the addition of propane between the front location and near the end of the reaction zone to provide constant composition entering the LIF detection cell (like the OH CIMS technique), (3) test various diameters and lengths of reaction, and various materials for the reactor walls, and (4) examine impacts of flow rate on mixing and CMR performance? Such tests, if performed, should be discussed in the paper. If not performed, their possible impacts should be presented

**Response:** We understand the remark and we have extended the discussion in the manuscript on why this particular CMR was used. The CMR needs to be implementable on both the LIF system permanently mounted at the SAPHIR chamber and the LIF system used in field campaigns and therefore the best compromise system was developed. In answer to the points raised by the reviewer, (1), a lower pressure in the CMR would indeed facilitate the mixing, but at the same time it would increase wall losses of OH radicals and would require a much larger concentration of scavenger. One thing that we feel was not clearly explained in the manuscript by reading the comments of the reviewer and has been now revised is that the goal with the CMR is to avoid scavenging OH radicals in the detection cell of the instrument. Therefore, having a larger concentration in the CMR is not favorable. In addition, to reduce the pressure in the CMR, a nozzle would be needed with additional OH radical losses which would reduce further the sensitivity of the instrument. As mentioned above, the goal with the CMR is to scavenge a known large (90 to 95%) of the atmospheric OH within the CMR avoiding that a significant fraction of OH radicals are scavenged in the detection cell. So, there is no advantage for switching the addition of propane between front and end location of the reaction zone (point (2)). To answer to point (3), the tube diameter was a good compromised between allowing for a reasonable reaction time at the flow rate chosen to happen and avoid the loss of a large fraction of OH radicals on the wall. Instrumental constrains (detection cell dimension and calibration source) also limit the diameter of the tube.

We included the result of transmission tests with Teflon and SilcoNert materials for the flow tube wall showing a better transmission of the OH radicals when using Teflon 64% as compared to 53% with SilcoNert. The flow rate within the CMR is somehow limited by the deployment in the SAPHIR chamber which does not allow for a large flow to be pulled by a single instrument and by the calibration source which provides at max 24 slpm. In addition, decreasing the flow would increase the reaction time but at the expenses of the OH radical transmission (see Figure S1). Therefore, the flow rate was kept at 21.1 slpm which resulted in 18ms residence time. The good reproducibility of the scavenging efficiency among one year (see section 3.4) and the good agreement between OH LIF and OH DOAS (see section 3.4 and 4.2.2) gave confidence that this design allows us to perform high quality OH radical measurements.

Figure S1 was included in the SI and the following text was added to section 3.1.

'The wall loss of OH radicals for a flow rate in the CMR between 12 and 21.1 slpm was investigated (Figure S1). As the flow rate increases from 12 slpm to 15 slpm, the OH transmission also increases from ~ 58 to 65%) due to the shorter residence time. Further increase of the flow rate does not improve the transmission efficiency which reaches a plateau of ~ 66% for flows above 16 slpm. To minimize the

possible effect of secondary chemistry, the CMR was operated with the fastest flow rate achievable (21.1 slpm), which corresponds to a transmission a transmission $\beta_{N2}^0$ of 0.64'

[Figure]

**Figure S1:** OH radical transmission efficiency measured while varying the flow rate in the CMR.

**Comment #64**: While some instruments have apparently not seen significant interferences beyond the impact of ozone photolysis by the probe laser, is it possible that it is because of the atmospheric environments in which the tests were done. Indeed, even in the present study, there doesn't appear to be a significant impact of biogenic VOCs on OH reactivity. Previous studies have indicated that forested environments, presumable with significant BVOC emission, have the largest measured interferences. This possibility should be added to discussion, including in the middle of page 3.

**Response:** We agreed with the reviewer that the environment where the measurements were performed were characterized by lower BVOCs emission compared to previous studies in forested area. We extended the discussion including what observed during the Wangdu campaign (Tan et al., 2017) where the CMR was used for the first time and where higher isoprene (up to 4 ppbv) was measured.

In Page 3, line 21, we added following text.

'Although these studies were not performed in forested environments with large BVOC emissions as some of the studies where a large interference was observed, during the summer campaign in Wangdu (Tan et al., 2017) and Beijing (Woodward-Massey et al., 2020), high isoprene concentrations (up to 3 ppbv and 7.9 ppbv, respectively) did not seem to perturb the level of the interference. These field campaign results are consistent with the results of laboratory and/or chamber studies (Fuchs et al., 2016; Woodward-Massey et al., 2020) which found insignificant interference from the ozonolysis of BVOCs under atmospheric conditions.'

Page 17, line 38 - 40 was changed as following text.

'Similar to what is observed during JULIAC, no significant unknown OH interference signals were detected during the field experiments with the exception of the campaign in Wangdu (Tan et al., 2017) where relatively high level of isoprene (up to 3 ppbv) was observed. Here, unexplained OH signals equivalent to $(0.5 - 1) \times 10^6$ cm$^{-3}$ were found with an experimental $1\sigma$ uncertainty of $0.5 \times 10^6$ cm$^{-3}$. Around noontime, these signals were less than 10 % of the ambient OH signals. However, the magnitude of the interference was small compared to the field observations with other LIF instruments in forested environments (Mao et al., 2012; Hens et al., 2014; Novelli et al., 2014a; Feiner et al., 2016; Lew et al., 2020).'

**Comment #65**: This reviewer found the use the various Greek symbols ($\alpha$, $\beta$) with multiple superscripts and subscripts not clearly defined. While Figure 2 was helpful in this regard, not all of the symbols are included. It might be easier for the reader if the symbols were more mnemonic. In other words, use T for transmission of radicals (as on page 12) throughout the paper with a clear definition (the fraction of radicals successfully transported through the CMR), and another symbol (perhaps R) for the reverse of transmission, which is referred to as the residual factor (suggest a different term) and definition (the fraction of radicals lost during transport through the CMR). This way it is easier for the reader to remember what the various symbols stand for. This reviewer suggests the authors consider a table with the symbols used in the paper to aid the reader.

**Response:** We agreed with the reviewer that a table is needed to help following the many formulas in the manuscript. We have added the table in Appendix (Table A1).

Regarding the change of the names, we think that different people from different countries might associate letters to different meanings and therefore do not see the advantage of the changes suggested. We did though clarify all the definitions.

**Table A1.** Specification of CMR parameters used in this study.

[revised manuscript text omitted]

a Variable

b Dimensionless

**Comment #66**: This reviewer tried to reproduce the plots in Figure 7 based on information in the paper. It was not possible arrive at the exactly the same figures. Suggest that you carefully check the calculations (perhaps by at least two people independently) to ensure that these calculations are correct.

**Response:** As the reviewer suggested, all calculations were carefully checked by two people independently.

**Specific comments.**

**Comment #67**: Page 1, lines 11 & 12. Suggest replacing the word "how" used twice in this first sentence. For example, "…are essential to investigate mechanisms for oxidation and transformation of trace gases..." and "…and processes leading to the formation of secondary pollutants…"

**Response:** Done.

**Comment #68**: Page 1, line14. The use of chemical modulation has been utilized in most LIF instruments "recently". Somewhere, if possible, give a reference to the first suggestion, or a recent suggestion, that this is how such measurements should be performed.

**Response:** Done.

**Comment #69**: Page 1, line 17. While this reviewer agrees that this paper describes the application and characterization of the CMR, it is not obvious that it has been validated. Suggest adding text, perhaps near the end of the Introduction that demonstrates the CMR has been validated. Later, on page 3, the words "introduced, described, and characterized" are used, which might be better here as well.

**Response:** Following the comment from the reviewer we have removed the word "validation" from the abstract.

**Comment #70**: Page 1, line 20. Suggest rewording "It allowed performing OH…". Perhaps "enabled" instead of "allowed" or a reorganization of the sentence.

**Response:** Done.

**Comment #71**: Page 1, line 22. Suggest removing "A" to read "Good agreement was obtained…".

**Response:** Done.

**Comment #72**: Page 1, line 22. Suggest a "-" or "/" or "vs" in "LIF-DOAS".

**Response:** Done.

**Comment #73**: Page 1, lines 22-23. This reviewer takes issue that good agreement between LIF and DOAS confirms that the CMR provides interference-free OH measurements. It depends on whether the comparison was performed in conditions that cause interference, and it assumes that there is no possibility of interference in DOAS. While the latter seems unlikely, it cannot be ruled out completely. In fact, the statement on line 30, indicates that the situation of JULIAC did not include unexplained interference. Thus, it might not be suitable to use the setup described in a forested environment with large BVOC emissions. This should be discussed somewhere in the paper, and perhaps the case made not quite to strong in the abstract.

**Response:** We feel as there was a misunderstanding with the meaning of interference-free OH measurement and we clarified that in section 4.2.3. Independently on the environment, the LIF operated by the Forschungszentrum Jülich is known to be affected by a small ($3.4 \times 10^5$ cm$^{-3}$ per 50 ppbv of $O_3$ and 1 % water mixing ratio) ozone interference (Holland et al., 2003) (section 3.5.1) which is accounted and corrected for. When the CMR is added to the LIF, this correction is not needed as the CMR allows for the separation of the atmospheric OH radical, scavenged within the CMR, and any interference (which includes also the known ozone interference). Although it is true that a very large interference (larger than the atmospheric OH radical) would result in a more scattered atmospheric OH signal (as observed in the study by Novelli et al. (2014)) that would still constitute an interference-free OH radical signal. In addition, as highlighted in this study, the implementation of the CMR could introduce fictitious interferences if the chemistry within the CMR is not correctly accounted for. We have introduced a model to account for this and the good comparison with the OH measured by the DOAS confirms we can account for that. So, we think we provide interference free OH radical with the CMR LIF.

According to the suggestion of the reviewer we have underlined that what described in this study refers to a specific set of chemical and meteorological conditions.

'...chemical modulation system of the FZJ-LIF instrument has been  suitable for measurement of interference-free OH concentrations under the conditions of the JULIAC campaign (rural environment).'

**Comment #74**: Page 1, lines 27-28. The description of the medium diurnal variation of the interference could be misleading. I think you mean that the maximum value of 0.8 x 106 occurs during daytime, and the minimum value occurs at night, but it could be interpreted other ways. Suggest rewording the sentence.

**Response:** Done.

**Comment #75**: Page 1, lines 33-37. The models described in the paper do not seem to be chemical kinetic models (nor kinetic chemical models). Also, the language could be improved in these sentences. For example, "A kinetical chemical model of the chemical modulation reactor was developed…". Suggest substituting a different word to avoid using "chemical" twice or rearranging the sentence. In the next sentence, "reactions" and "reactor' are used close together that makes the sentence a bit awkward. This reviewer prefers "photolytic" to "photolytical", but which one the author uses is their choice.

**Response:** Done.

**Comment #76**: Page 1, lines 40-41. This conclusion that chemical modulation can be subject to interferences while attempting to correct for other interferences is an important finding. It should be clearly stressed (maybe with a bit more detail) here and elsewhere in the paper.

**Response:** We agree with the comment from the reviewer. Page 1, line 40-41 was changed as following:

'However, in environments with high OH reactivities, such as in a rain forest or megacity, the expected perturbation in the currently used chemical modulation reactor could be large (more than a factor of 2) . Such perturbations need to be carefully investigated and corrected for the proper evaluation of OH concentrations when applying chemical scavenging.'

**Comment #77**: Page 2, line 14. The Stone et al., 2012 reference discussing OH measurements is good, but now a bit out of date. Suggest adding one or two more recent references.

**Response:** Done.

**Comment #78**: Page 2, lines15-16. Suggest rewording part of this sentence "…which helped in the investigation of the atmospheric OH radical budget."

**Response:** Done.

**Comment #79**: Page 2, lines 28-29. Suggest rewording "…in atmospheric chemical models increases the predicted concentrations of OH considerably for some conditions, for example…"

**Response:** Done.

**Comment #80**: Page 2, line 33. It might make sense to add discussion of measurements of OH by other techniques and the relationship between measurements and models. There is indication, for example, that CIMS could be prone to interferences but it is believed that they are accounted for by the way propane is used in the inlet.

**Response:** We agree with the comment from the reviewer and the change the text as following:

'Chemical modulation in instruments measuring OH,  is also commonly used by CIMS which measure OH by addition of $SO_2$ in form of $HSO_4^-$ by ionization with $NO_3^-$ (Eisele and Tanner, 1991; Tanner et al., 1997; Berresheim et al., 2000; Mauldin et al., 2010). It involves periodic scavenging of the ambient OH by addition of a reactant (propane or hexafluoropropene) before the air enters the detection cell'

In addition, we added a reference to the study by Wolfe et al. (2014) in Page 2, line 32.

'In most of  field campaigns described above, the OH radical was measured by gas expansion of ambient air into a low-pressure volume, where the radicals are detected by laser-induced fluorescence (LIF) at 308nm, except for the study by Wolfe et al. (2014) where the OH radical was measured by chemical ionization mass spectrometers (CIMS).'

**Comment #81**: Page 2, line 37. Suggest adding one or more reference to "Previous studies" that are mentioned, again including measurements of from CIMS as well as LIF.

**Response:** The possible OH interference is different for LIF and CIMS. Here, we want to focus on the OH interference for the LIF instrument.

**Comment #82**: Page 3, lines 1-2. The Hofzumahaus and Heard reference does not suggest that use of chemical modulation necessarily leads to OH measurements that are interference free. Suggest rewording this sentence.

**Response:** Done.

'In accordance with a recommendation from the International HOx workshop 2015 (Hofzumahaus and Heard, 2016), the majority of LIF instruments now apply chemical modulation in order to correct possible interferences .'

**Comment #83**: Page 3, lines 10-11. Again, while some instruments see large interferences, it may because of where they were deployed rather than the configuration of the instrument (or a combination). Suggest rewording and/or adding text to make this clear.

**Response:** Done.

'Several LIF instruments applying chemical modulation (MPIC, PSU, and IU in Table 1) have shown relatively large OH interferences which seem to depend on the chemical conditions of the sampled air as well as on specific instrument characteristics (e.g. inlet size and shape or whether the detection cell is single-path or multiple path) and/or from the combination of both.'

**Comment #84**: Page 3, line 8. Again, the term "interference-free" is used. This has not been unequivocally demonstrated. Indeed, the FZJ use of the CMR leads to a reduction of the interference, but not complete removal as shown later in the paper.

**Response:** Please refer to our answer to the Comment #73 (Page 1, lines 22-23).

**Comment #85**: Page 3, lines 16-17. Suggest rewording "…but nighttime observations were similar to those found in forested environments."

**Response:** Done.

**Comment #86**: Page 3, line 24. Suggest changing "avoid" to "minimize".

**Response:** Done.

**Comment #87**: Page 3, line 30. Suggest "…is developed that provides estimates of the possible interferences…".

**Response:** Done.

**Comment #88**: Page 3, line 36. Suggest "…a large range of chemical…".

**Response:** Done.

**Comment #89**: Page 4, lines 5-6. Suggest rewording "…the excitation wavelength is modulated on and off the peak of the OH absorption line…".

**Response:** Done.

**Comment #90**: Page 4, line 8. Could you describe why the timing of on-line and off-line signal measurements are not the same? Normally, optimum signal to noise is achieved when they are the same, unless the noise of the background (off-line) is very small.

**Response:** The timing of on-line and off-line are different as the measurement of a short-lived species (e.g. OH radicals) requires the highest time resolution that one can possibly achieve. In addition, the noise on the background is very small.

**Comment #91**: Page 4, line 13. Suggest adding text to indicate that the intensity of the mercury lamp radiation must be known, and is determined by actinometry.

**Response:** We added the following text.

'The intensity of the light is determined by actinometry and it is measured by a calibrated photo-diode.'

**Comment #92**: Page 4, line 17. Suggest rewording "Hydroxyl radicals originating not from ambient air but formed within the detection cell…".

**Response:** Done.

**Comment #93**: Page 4, line 19. The physical location of the CMR is said to be "on top". While this may be true, up or down do not really apply to gas flows. Suggest using "in front" instead.

**Response:** Done.

**Comment #94**: Page 4, line 20 and following. The dimensions and details of the CMR are described, but it doesn't say why these parameters were selected. Suggest adding text describing the method by which the current design was arrived at.

**Response:** Please refer to our answer to the Comment #63.

**Comment #95**: Page 4, line 21. The injectors are 1/8" OD tubes, with 50 μm ID, correct? Suggest rewording.

**Response:** Corrected.

**Comment #96**: Page 4, line 25. For CIMS measurements, it was found that the purity of the propane is very important. Even small amounts of impurities can cause problems. Please give the purity of the propane use to make the propane/N2 mixture. Indicate whether experiments with different sources (vendors) of propane have been performed. This is very important.

**Response:** We are using a high purity (purity$\geq$99.95%) propane mixed with a high purity $N_2$ purity>99.999%). The possible impurities of the mixture (5% propane in $N_2$) would be smaller than what one could observe for a pure propane cylinder. From our regular scavenging efficiency tests with different mixture cylinder, we did not observe a significant change of the experimental residual factor. Therefore, we do not expect a significant impact from the impurities of the propane.

According to the reviewer's suggestion, we changed Page4, line 25 to be '…(Airliquid, Propane, purity>99.95%, 5.0±0.1% mixture in nitrogen, purity>99.999%)…'

**Comment #97**: Page 4, line 30. Why are the measurement cycles so much longer with the CMR in place?

**Response:** The measurement time includes the addition of propane (135 s), followed by another period without propane injection (135 s). This cycle is set to provide enough time to flush the injected propane (in $N_2$ mode) and achieve homogeneous mixing of propane (in scavenger mode).

**Comment #98**: Page 4, line 33. Why do you do on-line and off-line measurements while using the CMR? This should not be necessary in normal measurement mode (but perhaps useful when doing tests). Equations 4 and 5 show that it isn't necessary to measure Si.

**Response:** The reviewer raises an important point. Indeed, we do not need to do wavelength modulation as it cancels out when we calculate the difference of $N_2$ and SC modes. However, the wavelength modulation is needed for the measurement of $HO_2$ and $RO_2$ radicals as the laser beam passes through the ROx, OH and, HOx cells in sequence.

**Comment #99**: Page 5, line 8. Suggest "…the OH cell with the CMR in place…".

**Response:** Done.

**Comment #100**: Page 5, lines 10-13. This conclusion may not be valid if there are contaminants in the propane.

**Response:** Please refer to our answer to the Comment #96 (Page 4, lines 25).

**Comment #101**: Page 5, line 17. The DOAS does not require calibration, but it does depend on knowledge of the absorption cross section for the pressure and temperature conditions of the measurement, and the absorption path length. Please add this information to this section.

**Response:** We extended the description of the DOAS instrument by adding the following text.

'The calibration-free differential optical absorption spectroscopy (DOAS) instrument (White cell, absorption path length: 2.2 km) provided…'

'The pressure and temperature dependence of the OH absorption cross section has been discussed in detail in the study by Dorn et al., 1995. Within the natural variance of the atmospheric pressure and a temperature interval of ±20K around room temperature the OH cross section changes less than 2%.'

**Comment #102**: Page 5, line 22. It is probably better to say "…experiments an NO concentration…", since the reader is likely to read "an en oh" rather than "a nitric oxide".

**Response:** Done.

**Comment #103**: Page 6, line 1-2. Suggest "…which has a light transmission of greater than 0.8 over the complete solar spectral range…".

**Response:** Done.

**Comment #104**: Page 6, line 3. Suggest "…to prevent contamination from…".

**Response:** Done.

**Comment #105**: Page 6, line 6. The beginning of this section says that comprehensive tests were performed, but that is not clear from what is written. Did you do any tests with biogenic VOCs and ozone added to synthetic air to see if other interferences showed up? How about NO3 oxidation experiments of BVOCs? These are situations that could produce additional interferences and should be investigated and discussed.

**Response:** The tests the reviewer suggest have already been performed on the system described in this study and were extensively discussed in a previous publication (Fuchs et al., 2016) which was not adequately cited. As mentioned in the response to the comment #64, we extended the introduction in Page 3, line 21.

In addition, we changed page 5 line 33-34 as following;

'Comprehensive tests of the LIF-CMR instrument with synthetic (Section 3) and ambient air (Section 4) were performed in the atmosphere simulation chamber SAPHIR on the campus of Forschungszentrum Jülich, Germany.'

**Comment #106**: Page 6, line 9. Suggest "…measurements in SAPHIR during which ambient air was…".

**Response:** Done.

**Comment #107**: Page 6, line 11. Suggest "…located close to the city of Julich."

**Response:** Done.

**Comment #108**: Page 6, line 8-25. It is not clear why the experiments were performed in the SAPHIR chamber rather than just by sampling ambient air. Please add a sentence or two of explanation.

**Response:** We added below text in page 6 line 15.

'This new inlet system for SAPHIR enable all instruments to measure the same air composition without the perturbation of steady state conditions by local emitters.'

As we described in the response to the review's comment #63. The instrument is permanently mounted to the SAPHIR chamber. To clarify this we added below text in page 5 line 34.

'…on the campus of Forschungszentrum Jülich, Germany as the instrument is permanently mounted at the SAPHIR chamber.'

Lastly, we changed the text below at page 6 line 4.

'

The SAPHIR chamber is an ideal tool to test and characterize instruments for the measurement of atmospheric trace gases, as it was shown for the measurements of OH (Schlosser et al., 2007; Schlosser et al., 2009; Fuchs et al., 2012a), $HO_2$ (Fuchs et al., 2010b), $NO_2$ (Fuchs et al., 2010a), $NO_3$ (Dorn et al., 2013), $N_2O_5$ (Fuchs et al., 2012b), and OH reactivity (Fuchs et al., 2017). In this study, the chamber was used for

experiments with synthetic air (Linde, purity: > 99.99990 %), to test the CMR for known interferences from ozone/water and $NO_3$ radicals (Section 3), and with ambient air (Section 4).'

**Comment #109**: Page 6, line 21. Suggest "The total OH reactivity…".

**Response:** Done.

**Comment #110**: Page 6, line 23. Suggest "range" instead of "suite".

**Response:** Done.

**Comment #111**: Page 6, line 24. JULIAC indeed allowed potential interferences to be investigated, but only within the range of conditions found in Julich. There are other conditions throughout the world that are different and could present other interferences. This should be stated either here or elsewhere in the paper.

**Response:** we changed line 24 to be '…for different scenarios (e.g., summer/winter, high/low NO and $O_3$, high/low humidity) in a rural environment (Table 3).'

**Comment #112**: Page 6, line 35. It is not clear why the term "laminar flow tube" is used immediately followed by stating that it exhibits plug flow conditions. While it is possible to experience approximate plug flow in a tube, this is usually found at low pressures (< a few mbar) and in the entrance region (before the full flow regime is established). It is likely with the calibrator to be due to the latter. It would be useful to state the Reynolds number and remind the reader of the flow regime, both in the calibrator and the CMR. Then indicate why the calibrator might exhibit plug flow.

**Response:** The plug flow was confirmed by the measurement of ozone produced in the flow tube of the radical source. It was found that the ozone concentration in the central flow differed only a few percent to the concentration measured at the side (Page 6, line 36-38).

'The length of the laminar flow tube (18.7 mm ID with a frit at the top, Reynolds number Re = 1920) of the radical source is 20 cm, resulting in plug flow condition that ensures a uniform distribution of OH and $O_3$ in the calibration gas.'

**Comment #113**: Page 7, line 1. Suggest "… (Reynold number, Re = 2800) …". This value of Re is actually in the transition regime (2300-2900), so the flow cannot be clearly classified as turbulent. Also, there is an transition region of up to tube diameters to establish the condition of a given Re. Finally, the injectors generate turbulent (to enhance mixing), so that has an effect as well.

**Response:** Line 1 at page 7 was rephrased as following,

'The flow in the entrance section is  in the transition regime between laminar and turbulent (Reynolds number, Re = 2800) but turbulence is further increased by the injectors which protrude approximately 4mm into the flow tube.'

**Comment #114**: Page 7, line 7. The experiment to measure transmission without the injectors is useful, but it would be interesting to see how it varies with flow rate and tube diameter. Were such experiments done?

**Response:** Please refer to our answer to the Comment #63 (general comment).

**Comment #115**: Page 7, line 9. Since the radical calibrator produces both OH and HO2, is there evidence of loss (or do you expect it from the kinetics) from the reaction of OH with HO2. It would be helpful to give the range of OH concentrations employed in the various tests performed, including this one.

**Response:** Within the calibration source, $5 \times 10^9 \text{cm}^{-3}$ of OH and $HO_2$ radicals are produced (for 0.8% water vapor mixing ratio). Given the rate coefficient for the reaction of OH with $HO_2$ ($1.1 \times 10^{-10} \text{ cm}^3 \text{ s}^{-1}$ at 298K), and the time between the radical production and the sampling (30 ms), a loss for both radicals of less than 3 % is expected.

According to the suggestion of the reviewer, we added the range of OH concentration from the calibration source in Page 4, line 13 to be '…a known amount of OH radicals (approximately $5.0 \times 10^9 \text{ cm}^{-3}$ at 0.8 % water vapor mixing ratio) by photolysis of water vapor at 185 nm'

**Comment #116**: Page 7, line 12. Suggest "The rate coefficient, kw, that was obtained can be used…"

**Response:** Done.

**Comment #117**: Page 7, line 20. Was the concentration of CO in the CMR varied to see whether it had an effect? Are you concerned that if there was surface conversion of HO2 to OH that could be released into the gasphase, it would be masked by the presence of CO. It seems that the HO2 loss experiment could be conducted without addition of CO, assuming reaction between HO2 and OH was not an issue (comment earlier).

**Response:** The CO concentration was not varied as the test was performed focusing on the determination of the $HO_2$ wall losses on the CMR. We are not familiar with a chemical process that would allow for conversion of $HO_2$ to OH radicals on the CMR flow tube. However, due to the low residence time in the CMR flow tube (17.8 ms) and the small loss rate of $HO_2$ radicals on walls ($14.5 \text{ s}^{-1}$) compared to the wall loss of OH ($33 \text{ s}^{-1}$), we do not think this process happens in a detectable amount. In addition, the good agreement between the OH radicals measured by the LIF with CMR and the DOAS confirms that we are not largely affected by this possible surface conversion**.**

**Comment #118**: Page 7, line 27. There is a comma at the beginning of the line. Suggest rewording the text before and after equation (11) so this is not needed.

**Response:** Done.

**Comment #119**: Page 7, line 33. Perhaps remind the reader that the following discussion is for clean air, in other words kOH = 0. You also might want to use the superscript 0 to indicate this in all the terms being presented. It is also not clear the superscripts "e" and "r" were defined on the previous page.

**Response:** As the review's suggestion, we added the superscript 0, for example, $\beta_{N2}^{r,0}$.

The following text was added at page 7 line 34.

'The same radical source operated with clean synthetic air was also used for the characterization test of scavenging efficiency.'

In addition, line 13 at page 7 was changed to be '...estimate the transmission of the entrance section ($\beta^{e,0}$),'

**Comment #120**: Page 8, line 3. Suggest "where ksc is the pseudo first-order rate constant for reaction between OH and propane."

**Response:** Done.

**Comment #121**: Page 8, line 5. Suggest "The fraction of ambient OH remaining that subsequently enters the detection cell…".

**Response:** Done.

**Comment #122**: Page 8, line 12. Suggest "…as expected from Equation 16, whose derivation assumes homogeneous mixing…".

**Response:** Done.

**Comment #123**: Page 8, line 27. This reviewer disagrees that further increases in the propane concentration gives only small improvements in α. Looking at the data rather than the red line, there is little indication that the curve is flattening. The y-axis on this plot is logarithmic with magnitude orders of magnitude. Perhaps four symbol diameters correspond to a factor of two change in the residual amount remaining. Because of the issues with too much propane in the detection cell, experiments should be performed with longer length reactor tubes. It may be that a 50% increase is sufficient to bring α close to zero. This reviewer suggests performing such tests before publication of this paper.

**Response:** We agree that it is hard to see the curve flattening with a logarithmic scale. With a non-logarithmic scale as in the figure below, it can be seen that a further increase of the propane concentration above 20 ppmv reduces alpha only by less than 3%.

[Figure]

In addition, as mentioned in our response to the Comment #63, we aim to scavenge 90-95% of OH radicals within the CMR, and a longer flow tube and therefore a longer residence time would result in higher OH wall losses.

**Comment #124**: Page 8, line 33. Is it also possible that ultraviolet light, either from the radical source or ambient sunlight enters the CMR and affects the conversion chemistry? This possibility should be discussed along with data collected using the apparatus.

**Response:** The experimental budget study of OH radical (not shown in this study) during JULIAC shows relatively small contribution of HONO and $O_3$ photolysis (~20% and ~3%, respectively) compared to the reaction of $HO_2$ with NO (~75%). If we assume the incoming number of photons to the CMR is 1/100 of the ambient level, OH production from the photolysis reaction is almost negligible. Therefore, we think that the assumption that no-photolytic chemistry happens in the flow tube is valid.

**Comment #125**: Page 9, Line 9. Suggest "As employed in the HO2 transmission tests…"

**Response:** Done.

**Comment #126**: Page 9, line 5-25. It appears to this reviewer that higher amounts of propane could be used in the CMR with minor impacts in the detection cell. For reasons that are not clear, a 3% effect is considered negligible, where a 5% effect is unacceptable. It is excellent that this range of propane levels was tested to measure this affect. This reviewer suggests such range for the scavenging efficiency tests.

**Response:** As highlighted in the answer to comment #63 and 123, the goal with the CMR is to scavenge a known large amount of OH radicals (90 to 95%) avoiding that a significant fraction of the OH radicals are scavenged in the detection cell. Therefore, we prefer to keep the optimized propane concentration at the value used within this study (19 ppmv)

We changed Page 9, line 17 to be 'A small  amount of (3±2) % of…'

**Comment #127**: Page 9, line 33. "photolytical" is used again.

**Response:** Corrected with photolytic.

**Comment #128**: Page 9, line 35. Suggest "The instruments agreed to well within the combined 1σ accuracies…". Can you demonstrate this numerically, for example by providing the mean and standard deviation of the differences between the two measurements (or some other metric)? Page 10, line 1. When regression is mentioned throughout the paper, please indicate whether it is standard least squares or, as mentioned in one case, a bivariate technique such as fixexy in Press.

**Response:** we changed line 35-36 at Page 9 as following:

'The difference between the measured time series is on average $(0.29 \pm 0.9) \times 10^6$ cm$^{-3}$, which is less than 5% of the average measured OH concentrations between 08:30 and 12:00. Thus, the instruments agreed well within the combined  1σ accuracies of the LIF-CMR calibration (± 18%) and DOAS (± 6.5%), which confirms the LIF calibration.'

In addition, we changed Page 10, line 1 to be 'A linear regression (not shown) which considers the precision of both instruments using the algorithm Fitexy between…'

**Comment #129**: Page 10, line 8. This was mentioned before, but why is it necessary to do the on- and off-resonance measurements when using the CMR?

**Response:** Please refer to our answer to the Comment #98 (Page 4, line 33).

**Comment #130**: Page 10, line 15. Suggest "…when air pollutants influence the chemistry in…"

**Response:** Done.

**Comment #131**: Page 10, line 15. Indicate the level of uncertainty (1σ?) for the accuracy statement (here and other places in the paper).

**Response:** Done

In addition, line 12 at page 10 was changed to the following 'The 1σ accuracy of OH….'

**Comment #132**: Page 10, line 19. Suggest "…known interferences (ozone photolysis and NO3 radicals)…"

**Response:** Done.

**Comment #133**: Page 10. Line 21. Suggest "…interferences from other sources." This is because there could be sources other than chemical compounds, perhaps.

**Response:** Done.

**Comment #134**: Page 10, line 28. Suggest "…and water vapour were varied up to 450 ppbv and 1.8%..."

**Response:** Done.

**Comment #135**: Page 10, line 34. It is stated that the ozone photolysis interference is linear with the product of ozone and water vapor, but the parameterization in Figure 13 is not linear. Please give more information.

**Response:** We obtain the slope of the $O_3/H_2O$ interference by linear regression from the experiments shown in figure 5, and then we applied it to the JULIAC data shown in figure 13.

The slight curvature in figure 13 is due to water quenching. The O3/H2O interference correction is done to the instrument signal (Cts mW$^{-1}$ s$^{-1}$). The corrected signal is then converted to the OH concentration applying the determined instrument sensitivity which must be corrected for water quenching. With increasing water values, the quenching increases resulting in the observed deviation from linearity. More information was added in section 4.2.3.

**Comment #136**: Page 10, lines 30-32. Why are the measurement techniques in this experiment different from Table 2?

**Response:** Corrected

**Comment #137**: Page 11, line 8. Suggest "…chamber using the thermal decomposition of N2O5 added from a condensed source which produces NO3 radicals. Indicate measurement technique for NO3 in these experiments. Can you explain the large amount of scatter in the Figure 6 plot?

**Response:** line 8 was changed following the suggestion from the reviewer.

During the NO$_3$ experiment, the sensitivity for OH was not optimal ($<0.1\times10^{-6}$ counts mW$^{-1}$ s) as compared to the sensitivity during the JULIAC campaign ($0.1\sim0.14\times10^{-6}$ counts mW$^{-1}$ s). This caused the large scatter for this experiment.

**Comment #138**: Page 11, line 19. Suggest "This section describes studies of the application….".

**Response:** Done.

**Comment #139**: Page 11, line 33. As mentioned earlier, how do you know that there is no photolysis in the CMR? Suggest not using "inner darkness" but use a more precise term.

**Response:** Please refer to our answer to the Comment #124 (Page 8, lines 33). The sentence was rephrased as following: 'When atmospheric OH radicals enter the  CMR, photolysis will  be significantly slowed (P$_{h\nu}\approx0$),…'

**Comment #140**: Page 12, line 2. Suggest "…is the dominate non-photolytic source…". (This reviewer agrees that photolytic is the correct term here).

**Response:** Done.

**Comment #141**: Page 12, line 6. It should be recognized that in the normal atmosphere, the ratio of HO2 to OH can be quite large, perhaps 100 or much more. In that case, a conversion of a small fraction of HO2 to OH could greatly change the amount. Suggest reworking the discussion to reflect this.

**Response:** We feel there was a misunderstanding and we have clarified the text. At line 6 of page 12 we wanted to highlight how the concentration of HO$_2$ is not changing within the flow tube of the CMR as even for a high concentration of NO (30 ppbv) only 10% of HO$_2$ will have reacted away within the short residence time.

**Comment #142**: Page 12, line 8. It is not true that NO is determined primarily NO reaction with ozone. This could be true, but there are also situations when the reaction of NO with peroxy radicals (HO2 and RO2) is comparable or even larger than NO+O3. Suggest rewording this sentence.

**Response:** We changed line 8 to be 'NO has an even longer chemical lifetime than HO$_2$, which is usually determined by the reaction of NO with O$_3$ and peroxy radicals (HO$_2$ and RO$_2$)'

**Comment #143**: Page 12. As a reader, I was confused by this section of equations. It was not clear how the terms and symbols used earlier applied to this section. This is why this reviewer suggests a table or figure to present all the various symbols and terms used. It is also not clear why there are two terms for transmission, β and T. With the correct corresponding subscripts and superscripts, one should be enough.

**Response:** We agreed with the point raised by the reviewer and we replaced T with β. Also, as mentioned in the response to comment #65, we added a table summarizing all the term used in the formulas in the Appendix (Table A1).

**Comment #144**: Page 13, line 1. It should be noted that kOH is never 0. It can be as low as 0.5 s-1 for very clean air, but all air has methane and CO.

**Response:** We rephrased line 1 to be '… up to 100 s$^{-1}$,…'

**Comment #145**: Page 13, line 5. Suggest reminding the reader why the minimum value is 0.042. Also, after using β values up to this point, why are the plots of α values?

**Response:** We rephrased line 5 as following 'α remains constant (same as the experimental $\alpha_0 = 0.042$)'

α is needed to calculate the ambient OH signal and the total OH interference signal with the CMR (Equation 4 and 5), while β is needed when the signal is converted to a concentration. As the order of the evaluation process, we described first α values. However, $\beta^e$, $\beta^r_{N2}$, and $\beta^r_{sc}$ need to be introduced and discussed as they are used to derive α (Equation 15).

**Comment #146**: Page 13, first paragraph. Mention that Figure 7b will be discussed later, in Section xx.

**Response:** we added to page 13 line 11, 'The effect of the change of transmission efficiency (β, Fig. 7b) will be shown in following Section 4.1.4.'

**Comment #147**: Page 13, line 8. Suggest "…the production Pd exceeds the photolysis-based OH production…"

**Response:** Done.

**Comment #148**: Page 13, line 13. Suggest removing comma "…would be obtained if the influence…"

**Response:** Done.

**Comment #149**: Page 13, line 15. Suggest "…signal Si* is similarly:"

**Response:** Done.

**Comment #150**: Page 13, line 18. Instead of "all kinds of" give the range of values over which it applies. It could be for the full range of values studied.

**Response:** We changed line 18 to be '…for  the range between 0 – 1 of…'

**Comment #151**: Page 13, line 28. Suggest "The result presented in Figure 8a is very similar to the homogeneous mixing case shown in Figure 7a."

**Response:** Done.

**Comment #152**: Page 13 line 36 and page 14, line 1. What is meant by "uncertainty of the model". Suggest being clearer here or remove this assertion. It seems that real differences are expected between the two models

**Response:** We removed the assertion.

**Comment #153**: Page 14, line 6. It is not clear why the inverse of the term in equation 31 is plotted in Figure 7b.

**Response:** The inverse of the transmission correction was used to see the impact of the corrections (α and β) on the OH concentrations compared to the OH concentration without the consideration of the corrections ([OH]*/[OH]), in Figure 10 (d)).

**Comment #154**: Page 14, line 12-13. This is a very important point that needs to be clearly stated in the abstract and the summary.

**Response:** we changed page 18 line 40 to be '…produce serious perturbations on the transmission and the scavenging efficiency of  the CMR…'

**Comment #155**: Page 14, line 17. Suggest "…OH production continues when the air enters…".

**Response:** Done.

**Comment #156**: Page 14, line 18. Suggest "…When kOH is much larger than…".

**Response:** Done.

**Comment #157**: Page 14, line 21. Suggest "…causes a relatively small…".

**Response:** Done.

**Comment #158**: Page 14, line 29. Suggest "…measured by DOAS for some of the JULIAC periods…".

**Response:** Done.

**Comment #159**: Page 14, lines 30-31. Suggest "…the campaign afforded the opportunity to test the chemical…".

**Response:** Done.

**Comment #160**: Page 14, line 32-33. Suggest "For only a short time between 1 to 11 February 2019, OH detection was done…".

**Response:** Done.

**Comment #161**: Page 15, line 12. Suggest "…is only weakly dependent on kOH and…".

**Response:** Done.

**Comment #162**: Page 15, line 14. What does "…and otherwise add up" mean? This needs some more explanation.

**Response:** We extended the explanation of the Figure 10d in section 4.2.1.

**Comment #163**: Page 15, line 14. It seems that the average correction is not so important, but rather the size and frequency of larger corrections. If the corrections are always small, there is no need to use the CMR. If they are sometimes big, and you don't know when, then it is important to have the CMR capability. Suggest rewording this part

**Response:** We agree with the reviewer. Since it is relatively hard to predict if measurements in one specific environment result in large interferences, the best approach is to always use the CMR with the correction. What we wanted to highlight is that, specifically for this dataset, due to the relatively small OH reactivity observed, the additional correction due to the change in the OH concentration within the CMR is small. As

this correction requires availability of several additional chemical species that would reduce the OH radical dataset during the JULIAC, we decided not to apply the chemical correction for the dataset within this study. We have better emphasized this at the end of section 4.2.1.

**Comment #164**: Page 15, line 19. Is this better for "The combined OH summer dataset measured…"?

**Response:** Done.

**Comment #165**: Page 15, line23. Can the authors provide some metric to indicate that the two LIF measurements indeed agree within their combined uncertainties?

**Response:** Following the suggestion from the reviewer, we extended the discussion of the two OH measurements by DOAS and LIF in section 4.2.2. We also added Figure S2 and a description about the systemic and statistical noise from the LIF instrument.

[Figure]

**Figure S2:** Residuals of the OH concentrations measured by FZJ-LIF-CMR, obtained from the difference between the measured OH concentrations and the linear fit shown in Figure 12 plotted vs the OH concentrations measured by DOAS. Vertical bars denote the combined 1σ precision of the measured data points from the both instruments. Blue dashed lines represent the value of the standard deviation of the residuals.

**Comment #166**: Page 15, line 23. It should be noted that the differences between the two OH measurements were during the heatwave.

**Response:** Done

**Comment #167**: Page 15, line 24. Suggest "… systematically higher than the DOAS measurements by about 25%."

**Response:** Done.

**Comment #168**: Page 15, line 26. See earlier comment about regression and fixexy.

**Response:** Done.

**Comment #169**: Page 16, line 6. Is it meant to refer to equation 19 rather than equation 12?

**Response:** Corrected

**Comment #170**: Page 16, line 11. Has the limit of detection versus averaging time been explored? It would be good to calculate the Allan-Werle variance for the LIF-CMR instrument and discuss in this paper. The data in Figure 13 have hours of data included, so the uncertainties could potentially be much smaller.

**Response:** We realized that the description of the error bars in the figure caption was not correct. Indeed, the error bars shows the variability of data within binned $O_3 \times H_2O \times Laser$ Power range, not the uncertainties of the measured OH interference.

Therefore, we changed Figure 13 caption. '(error bars are 1σ errors)' to be '(error bars are standard deviation of the bin averaged data)'. The uncertainty of the data shown in Figure 13 is much smaller ($<2x10^5$ cm$^{-3}$) than the variability.

**Comment #171**: Figure 1. Some suggestions to the body of the paper also apply to the figure caption. Suggest "Schematic drawing" rather than Technical drawing. Suggest "in front" rather than "on top". Suggest providing the mixing ratio rather than "traces of propane".

**Response:** Done.

**Comment #172**: Possibly include other symbols in this figure (see suggestions made earlier).

**Response:** Please refer to our answer to the Comment #65 (Page 17, lines 38-40). We included all symbols in Table A1.

**Comment #173**: Figure 4. Suggest "The data shown are…".

**Response:** Done.

**Comment #174**: Figure 5. Suggest using the same x=axis on this figure and Figure 13.

**Response:** Done.

**Comment #175**: Figure 7. In the body of the paper, it would be good to describe why these various terms are being plotted. What are each of these factors meant to show to the reader?

**Response:** Done.

**Comment #176**: Figure 9. The values in the plots do not seem to agree with the measurement period averages in Table 3. For example, for winter compared to summer, the value of j(O1D) is about 10 times larger in summer, while NO is about 3 times higher in winter. This seems to indicate that the fraction of OH from photolysis should be about three times higher in summer than winter. But the figure seems to

indicate than the photolysis fraction is about twice as high in the winter as the summer. Suggest checking the calculations for this figure and Table 3.

**Response:** The ratios cannot be simply calculated by using only $j(O^1D)$ and NO. Even though NO is 3 times higher in winter, summer $HO_2$ concentration is 5~10 times higher than in winter. In addition, $O_3$ concentration is largely different between summer and winter.

To make table 3 and figure 9 comparable, we added the calculated $P_d$ and $P_{hv}$ values to table 3.

**Comment #177**: Figure 13. Suggest giving the equation number for the parameterization. Adjust text as appropriate based on earlier comment.

**Response:** Done.

**Comment #178**: Table S1. Note that the detection limit depends on the averaging time (see earlier comment), so provide averaging time corresponding to the detection limit shown.

**Response:** Done.

**Comment #179**: Figures S1, S2 and S3. Suggest "Dashed lines denote midnight."

**Response:** Done.

**References**

[revised manuscript text omitted]